



# Chemical and dynamical identification of emission outflows during the HALO campaign EMeRGe in Europe and Asia

Eric Förster[1], Harald Bönisch[1], Marco Neumaier[1], Florian Obersteiner[1], Andreas Zahn[1], Andreas Hilboll[3,†], Anna B. Kalisz Hedegaard[2,3], Nikos Daskalakis[3], Alexandros Panagiotis Poulidis[3], Mihalis Vrekoussis[3,4,5], Michael Lichtenstern[2], and Peter Braesicke[1]

[1]KIT - Karlsruhe Institute of Technology, Institute of Meteorology and Climate Research, Karlsruhe, Germany
[2]DLR IPA - Deutsches Zentrum für Luft- und Raumfahrt, Institut für Physik der Atmosphäre, Oberpfaffenhofen, Germany
[3]IUP Bremen - Institut für Umweltphysik, Universität Bremen, Bremen, Germany
[4]Center for Marine Environmental Sciences (MARUM), University of Bremen, Bremen, Germany
[5]Energy, Environment and Water Research Centre, The Cyprus Institute (CyI), Nicosia, Cyprus
† deceased

*Correspondence to*: Eric Förster (eric.foerster@kit.edu) and Harald Bönisch (harald.boenisch@kit.edu)

**Abstract.** Worldwide the number of large urban agglomerations is steadily increasing. At a local scale, their emissions lead to air pollution, directly affecting people's health. On a global scale, their emissions lead to an increase of greenhouse gases, affecting climate. In this context, in 2017 and 2018, the airborne campaigns EMeRGe (Effect of Megacities on the transport and transformation of pollutants on the Regional to Global scales) investigated emissions of European and Asian major population centres (MPCs) to improve the understanding and predictability of pollution outflows. Here, we present two methods to identify and characterise emission outflows probed during EMeRGe. First, we use a set of volatile organic compounds (VOCs) as chemical tracers to characterise air-masses by specific source signals, i.e. benzene from anthropogenic emissions of targeted regions, acetonitrile from biomass burning (BB, primarily during EMeRGe-Asia) and isoprene from fresh biogenic signals (primarily during EMeRGe-Europe). Second, we attribute probed air-masses to source regions and estimate their individual contribution by constructing and applying a simple emission uptake scheme for the boundary layer which combines FLEXTRA back-trajectories and EDGAR carbon monoxide (CO) emission rates (acronyms are provided in Appendix A). During EMeRGe-Europe, we identified anthropogenic emission outflows from Northern Italy, Southern Great Britain, the Belgium-Netherlands-Ruhr area and the Iberian Peninsula. Additionally, our uptake scheme indicates significant long-range transport of emissions from the USA and Canada. During EMeRGe-Asia, the emission outflow is dominated by sources in China and Taiwan, with further emissions (mostly from BB) originating from Southeast Asia and India. Emissions of pre-selected MPC targets are identified in less than 20 % of the sampling time, due to restrictions in flight planning and constraints of the measurement platform itself. Still, EMeRGe combines in a unique way near- and far-field measurements, which show signatures of local and distant sources, transport and conversion fingerprints and complex emission mixtures. Our approach already provides a valuable classification and characterisation of the EMeRGe dataset, e.g. for BB and anthropogenic influence of potential source regions, and paves the way for a more comprehensive analysis and various model studies.



# 1 Introduction

Since the industrial revolution, the human population has strongly increased from ~1 to ~7.7 billion people in 2019 and is
predicted to increase to around 10 billion by 2050 (Goldewijk et al., 2017; United Nations, 2019). Cities have always been
centres of attraction, and since 2007 more people live in urban areas than in the countryside (United Nations, 2018). Today,
the largest urban agglomerations exceed more than 10 million inhabitants and are often referred to as megacities (Baklanov et
al., 2016; Folberth et al., 2015; Molina and Molina, 2004). The majority of megacities is still growing in population and
extension (United Nations, Department of Economic and Social Affairs, Population Division, 2018) and have even started
merging to form gigacities (Kulmala et al., 2021). According to a study by Folberth et al. (2012), 26 existing megacities (status
2012) alone were responsible for 12 % of the world's total annual $CO_2$ emissions (7 % for $CH_4$ emissions). In addition,
significant fractions of global emissions of air pollutants like $NO_x$ (4.6%), $SO_2$ (5.3%), Black Carbon (3.8%) and VOCs (4.8
%) are emitted from these 26 megacities.

In general, large urban agglomerations are considerable sources of long-lived greenhouse gases as well as short-lived
pollutants, and influence the atmosphere from local to global scales strongly affecting air quality and climate (Baklanov et al.,
2016; Zhu et al., 2012). Monitoring of emissions (and thus better quantifying pollution outflows of those areas) is required for
environmental policies and strategies to improve air quality, and also to refine and assess future climate projections. Previous
megacity projects focused e.g. on emissions of Mexico City in 2006 (MILAGRO, Molina et al., 2010), Paris in 2009-2010
(MEGAPOLI, http://megapoli.dmi.dk) and urban agglomerations in the Eastern Mediterranean, the Po Valley, the Benelux
area and the Pearl River Delta from 2008-2011  (CityZen, https://cordis.europa.eu/project/id/212095).

Existing knowledge gaps regarding megacity impacts on atmospheric composition involve inadequate characterisation and
prediction of pollution events and associated spatial patterns, including extent. In this respect, especially the interaction of
anthropogenic emissions with natural and biogenic emissions around urban agglomerations and further downwind is poorly
understood (Andrés Hernández et al., (2022) and references therein).

The international megacity campaign EMeRGe (Andrés Hernández et al., (2022), www.iup.uni-bremen.de/emerge)
investigates local, regional and inter-regional pollution transport originating from major population centres (MPCs) and their
effect on atmospheric chemistry and dynamics in Europe and Asia. During EMeRGe, dedicated airborne measurements with
an extensive set of instruments were performed with the German research aircraft HALO (High Altitude and LOng range
research aircraft, www.halo-spp.de) in July 2017 in Europe and in March/April 2018 in Asia. Compared to previous campaigns,
the HALO flights cover large areas with the goal to probe pollution outflows from multiple MPCs and to study their fate
(transport, chemical processing) in the sparsely in-situ probed domain between local and global scales. For a detailed
description and the aims of the EMeRGe campaign, we refer to Andrés Hernández et al., (2022). Hereafter, we will use the
term MPC to describe not only megacities but also urban agglomerations with several million inhabitants.

Within the long-range/large-scale measurement approach of EMeRGe, not all detected trace gas enhancements necessarily
originate from a single target MPC; other source regions could have contributed partly or even exclusively. These contributions





may come from nearby or have been transported over long distances, and are likely already chemically processed or diluted with background air. Thus, the challenge is to identify and subsequently to attribute emissions signatures in outflows, their contributing source regions and specific emission sources as well as the assessment of their transport and chemical processing. Here, we present a straightforward chemical and dynamical identification of probed emission outflows in terms of (potential)

emission sources and source regions. For the chemical identification of emission sources, we apply a multi-tracer approach that is based on the detection of different volatile organic compounds (VOCs) with a variety of atmospheric lifetimes (Atkinson, 2000) that are emitted by different anthropogenic and biogenic sources (de Gouw and Warneke, 2007). This "multi-tracer multi-lifetime approach" enables us to characterise air-masses in relation to source contributions and the stage of processing. Such an analysis is advantageous compared to the conventional approach of using a single tracer like carbon

monoxide (CO) that is emitted by anthropogenic sources and biomass burning (and that provides only one time information based on its lifetime of ~2 months).

This multi-tracer approach allows us to identify and distinguish emission signatures of biomass burning (BB), anthropogenic and biogenic origin. This is done based on concentration enhancements of different VOCs relative to their concentrations in background air. For the dynamical identification of source regions, we link back-trajectories of the Lagrangian FLEXible

TRAjectory model FLEXTRA (Stohl et al., 1995, 2005; Stohl and Seibert, 1998) and the EDGAR emission inventory (Emission Database for Global Atmospheric Research, https://data.jrc.ec.europa.eu/collection/edgar) to attribute anthropogenic emissions to (likely) source regions and estimate their contribution to the probed air-masses.

The joined use of both identification methods (measured VOC enhancements and inventory-based back-trajectories) allows the analysis and characterisation of emission fingerprints of the target MPCs and adjacent source regions. Within our work,

we focus on the following open questions:

a)  Which MPCs and regions have contributed with their emissions to the EMeRGe measurements?

b)  Which VOC-specific emission signatures are characteristic for the sampled target regions, in particular, what are the differences between Europe and Asia?

c)  How well does our multi-tracer approach work and what are its limitations?

d)  Which recommendations for future megacity campaigns can we derive from the results?

In section 2, we briefly describe the general observation strategy of EMeRGe. Section 3 explains our analysis methods, that is, the identification of a) emission signatures by chemical VOC tracers and b) air-mass origins with the use of a model-based approach where we inject specific amounts of CO into modelled air-masses (based on the EDGAR emission inventory and FLEXTRA back-trajectories) that arrive at our measurement locations. In section 4, we present our results and close in section

5 with a summary and conclusions.



## 2 Observation strategy

EMeRGe comprises 21 research flights (for detailed flight information, see Table S1 in the supplement) conducted with the German research aircraft HALO to probe emissions and photochemically processed air-masses at different altitudes and different distances downwind of European and Asian urban areas, with additional local ground-based and global satellite
measurements (Hernández et al., 2022).

To cover different types of areas with respect to population size, density and state of economic development, EMeRGe consists of two parts. In the first part, HALO performed seven research flights with 52 flight hours over Europe in July 2017 (EMeRGe-Europe) to investigate the urban area outflows of Madrid, Barcelona, London, Paris, Belgium-Netherlands-Ruhr (BeNeRu), Po Valley, Rome, and the region of Munich (where HALO is stationed). Favourable meteorological conditions for
photochemical processing under strong solar insolation influence prevailed in Southern Europe during the campaign period. Thus, heatwaves and fire events occurred there, whereas the passage of frontal systems accompanied by thunderstorms influenced only the northernmost flights. For more details, see  Hernández et al., (2022).

In the second part of EMeRGe, HALO performed 12 research flights (plus two European transfer flights) with 110 flight hours in March/April 2018 over Asia (EMeRGe-Asia) with its growing number of large megacities. HALO was stationed at Tainan
airport (Taiwan) to investigate the outflows of Taipei, Taiwan, Manila, Mainland China and South Japan. Spring is an inter-monsoon period with a) elevated levels of dust and biomass burning (BB), and thus characterized by mixing of different emission sources and their photochemical processing and b) storm and frontal systems which lead to a maximal outflow of the Asian continent to the Pacific Ocean (Cheng et al., 2014; Liu et al., 2003).

The overarching measurement approach of EMeRGe was to sample various kinds of air-masses on different scales, that is,
fresh emissions in the near-field (some 100 km) as well as processed, transported, and aged emissions in the far-field (> 500 km) of target regions. This concept is mirrored by the chosen flight altitudes shown in Fig. 1a, indicating that 65 % of measurements are performed below 3000 m altitude with expected near-field emission signatures and 35 % at higher altitudes with expected far-field signatures. The last contact of the sampled air-masses to the planetary boundary layer (PBL, retrieved from ERA5) in Fig. 1b shows, that 20 % of the measurements took place inside the PBL, 35 % had last contact to the PBL
within 10 days and 45 % more than 10 days, both in Europe and in Asia.

Inside the PBL, enhanced levels of pollutants originate mostly from recent emissions of distinct sources. In this emission stage, pollution plumes are still attributable to point sources. Here it should be noted, that we use the term "plume" only for small-scale emissions (e.g. from a factory or a local burning), for pollution enhancements of larger spatial extent (comprising e.g. various pollution plumes) we use the term polluted air-mass. Emissions leaving the PBL quickly start to mix with free
tropospheric air of different origin and chemical age, that is, chemically old or clean background air and likewise polluted air that is rich in decayed or diluted emissions of other pollution sources. During further transport, short-lived trace species will degrade (chemical transformation of timescales of up-to days), whereas long-lived species may be distributed on large scales (transported, stirred and mixed on timescales starting form hours to many days) and finally "become" background air.





The EMeRGe measurements cover these different transport, chemical aging, and mixing stages. Therefore, appropriate tools

are necessary to disentangle, characterise and attribute probed air-masses to source regions and identify relevant factors that

determined the resulting composition.

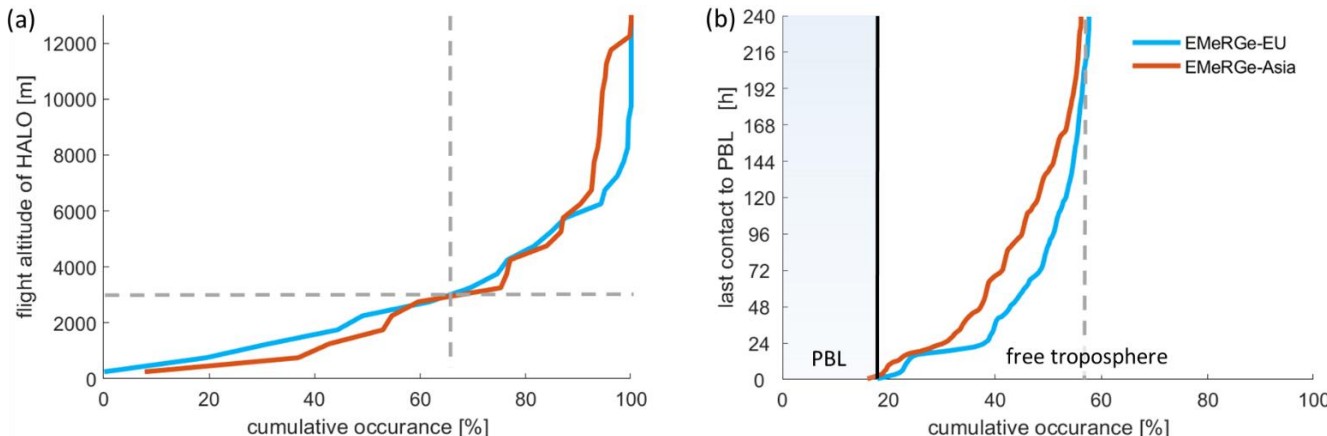

**Figure 1: (a) Cumulative occurrences of HALO flight altitudes during EMeRGe and (b) last contact of probed air-masses to**

**planetary boundary layer (PBL) air, based on 10-days FLEXTRA back-trajectories and ERA5 meteorological data.**

## 3 Methods

### 3.1 Identification of emission sources by chemical tracers

For emission source identification, we use source-specific chemical tracers from our set of VOC measurements. First, we

outline the measurement technique and the identification method by concentration enhancements above defined thresholds.

### 3.1.1 VOC measurements by PTR-MS

The technique of proton-transfer-reaction mass spectrometry (PTR-MS) was developed by Werner Lindinger and co-workers

(Lindinger et al., 1998, 2001) at the University of Innsbruck (Austria). In brief, an ion source produces hydronium ($H_3O^+$)

reagent ions from pure water vapour in a hollow cathode discharge. The $H_3O^+$ ions react with trace gases in the sample air

within a drift tube (reaction chamber). Proton transfer takes place, if the proton affinity of the target VOC is higher than that

of $H_2O$ (691 kJ mol$^{-1}$, NIST Chemistry WebBook, 2022). VOCH$^+$ and reagent ions are analysed in a quadrupole mass

spectrometer (QMS) or via time-of-flight measurements (ToF). Normalized VOCH$^+$/ $H_3O^+$ signals are converted to volume

mixing ratios (VMR) by calibration with external gas standards or estimated based on known proton transfer reaction rate

constants. The instrumental background (e.g. from impurities in the system) is usually determined by a catalytic converter and

subtracted from measurement signals (de Gouw et al., 2003). For a comprehensive description of the technique the reader is

referred to de Gouw and Warneke (2007) and Yuan et al. (2017).



In general, the vast majority of VOCs in the atmosphere is emitted by biogenic sources (Guenther et al., 1995; Sindelarova et al., 2014), thus from vegetation or biomass fires (Ciccioli et al., 2014). However, in and near urban areas, further anthropogenic sources like traffic and industry emissions dominate (Amodio et al., 2013). Additionally, the degradation of VOCs and other trace gases leads to secondary production of further VOCs, e.g. formaldehyde (de Gouw et al., 2009). Degradation of VOCs

is mostly due to the reaction with OH radicals and by photolysis (both occur during daytime) as well as the reaction with $NO_3$ radicals (in night-time). The combination and reactivity of these photochemical oxidation processes depend on the atmospheric conditions and on the chemical properties of the VOCs itself, which is why they possess a variety of life-times from hours to months (Atkinson, 2000).

VOC measurements during EMeRGe were performed with the HALO Karlsruhe Mass Spectrometer HKMS (Brito and Zahn,

2011), a custom-built PTR-MS (equipped with a QMS) which was continuously improved over the years (Fischbeck, 2017). Amongst its aircraft certification, the advantages compared to commercial instruments are its compactness, its low weight of 55 kg as well as its custom electronics and control software that allows adaptations and modifications to the scientific needs (Fischbeck, 2017). In the configuration for EMeRGe, the HKMS measured nine selected VOCs (Table 1) consecutively in a duty cycle of ~60 s (integration times of ~6 s per species) with lower detection limit (LOD) in the pptV-range. The instrumental

background is determined every 30 minutes for 5 minutes. The average $H_3O^+$ signal was $6 \times 10^6$ cps, the instrument was operated at a drift tube pressure of 2.3 mbar and an E/N value of ~142 Td. The sensitivities were determined in regular gas standard measurements. Data availability for EMeRGe is 95 % due to an instrument failure during flight EU-05.

**Table 1: VOCs measured during EMeRGe-Europe and -Asia, with tropospheric life-times t (as defined for a 12-h daytime average OH radical concentration of $2.0 \times 10^6$ molecules cm$^{-3}$; Atkinson, 2000)), lower detection limits (LOD, for EMeRGe-Asia), maximal**
**sensitivities S, mass spectrometric integrations times $\tau$ and the up-to four atmospheric main sources (highlighted by different colours).**

| Name | Protonated Mass [u] | Formula | t * | LOD [pptV] | S [ncps/ppbV] | $\tau$ [s] | Atmospheric main source(s) |
|---|---|---|---|---|---|---|---|
| isoprene | 69 | $C_5H_8$ | 2 h | 25 | 12.5 | 6.0 | Biogenic |
| $C_8$-aromatics | 107 | $C_8H_{10}$ | 6 h | 23 | 17.5 | 5.0 | anthropogenic, biomass burning (BB) |
| acetaldehyde | 45 | $C_2H_4O$ | 9 h | 72 | 25.0 | 4.0 | secondary production, anthropogenic |
| formaldehyde | 31 | $CH_2O$ | 1 day | 208 | 14.5 | 7.2 | secondary production, anthropogenic |
| toluene | 93 | $C_7H_8$ | 2 days | 17 | 17.6 | 7.2 | anthropogenic, BB |
| benzene | 79 | $C_6H_6$ | 10 days | 15 | 16.7 | 7.2 | anthropogenic, BB |
| methanol | 33 | $CH_3OH$ | 12 days | 644 | 12.3 | 6.0 | secondary prod., anthropogenic, biogenic, BB |
| acetone | 59 | $C_3H_6O$ | 2 months | 28 | 29.6 | 5.0 | secondary prod., BB, biogenic |
| acetonitrile | 42 | $C_2H_3N$ | 6 months | 22 | 27.3 | 4.0 | specific for BB |

**3.1.2 VOC-based identification of emission sources**

As indicated in Table 1, isoprene and acetonitrile are ideal tracers for fresh biogenic and BB sources, respectively, and benzene is a tracer marking anthropogenic signals when acetonitrile-identified BB signals are filtered out. In general, specific source

signals can be identified whenever certain tracer VMRs significantly exceed their atmospheric background levels. On that



basis, we can identify emission sources that have significantly contributed to measured air-mass composition when the respective VOC VMRs exceed the atmospheric background plus three times the (compound-specific) observational noise $\sigma$, as explained next.

We assess this observational noise $\sigma$ by smoothing the measured VOC VMRs with a Savitzky-Golay filter (Savitzky and 180 Golay, 1964) and taking the standard deviation of the residuals (details see supplement). The atmospheric background levels generally depend on the species' life time (Junge, 1974). Isoprene (emitted by forests and vegetation) is oxidised within some hours after emission and hence has a negligible atmospheric background. Benzene (emitted by gas-related traffic and petroleum processing industries) has a longer tropospheric lifetime of about ten days, but still shows very small free tropospheric background levels in the lowermost pptV-range. For both VOCs, we use therefore the instrumental LOD as lower limit. In 185 contrast, acetonitrile has a much longer tropospheric lifetime of about six months, and is thus distributed worldwide and hence shows a non-negligible tropospheric background. We infer the acetonitrile background level with the help of the most extensive in-situ data set collected with IAGOS-CARIBIC at altitudes of up-to 12 km (https://www.iagos.org/iagos-caribic/, Brenninkmeijer et al., 2007). Due to small BB emissions in winter, we select northern hemispheric winter seasons (DJFM, here for the four years 2012 to 2016) and determine a mean acetonitrile background level of 145 pptV (please see details in 190 the supplement).

Finally, we use the relevant background and threshold levels of acetonitrile, benzene and isoprene (listed in Table 2) to identify different emission signatures (summarised in Table 3). Overall, a quite complex and multi-faceted mix of chemical fingerprints characterising different emission sources emerges, with the following emission signatures:

- **Aged biomass burning (aged BB):** Elevated acetonitrile signals cannot only arise from recent burning events (together 195 with BB benzene signals, see next), but also distant events and long-range transport of air-masses with strongly reduced levels of short-lived benzene can contribute.

- **Biomass burning & benzene (BB & BEN):** An unambiguous source identification is challenging if both acetonitrile and benzene are enhanced. Such air-masses may originate exclusively from fresh BB events or may contain a mixture of anthropogenic and BB pollutants of varying ages.

200 - **Anthropogenic (AP):** Elevated benzene signals alone arise from fresh, some days old anthropogenic sources.

- **Only biogenic (only BIO):** Observed isoprene signals must originate from very recent biogenic emissions (due to the short life time of isoprene) and therefore denote recent contact with the PBL. This emission signature denotes only a biogenic signal without enhancements of acetonitrile and benzene. The above-mentioned three signatures are further partitioned into signals with and without biogenic influence (emission signatures II, Table 3).

205 - **Background (BG):** If none of the three VOC tracer is elevated, we consider the air-mass as background.



**Table 2: Upper tropospheric background levels of acetonitrile, benzene, isoprene (determined for ~11 km), limit of detection (LOD) and observational noise $\sigma$ used for inferring threshold levels to identify emission signatures during EMeRGe-Europe and -Asia (Note: Threshold limits are determined differently: background + $3\sigma$ for acetonitrile, LOD + $3\sigma$ for benzene and isoprene. Due to slightly changed sensitivities and instrumental backgrounds between 2017 and 2018, the LOD (depending on both) changed as well.)**

|  | parameters [pptV] | acetonitrile | benzene | isoprene |
|---|---|---|---|---|
| EMeRGe-Europe 2017 | upper tropospheric background | 145 | ~0 | ~0 |
|  | LOD | 37 | 29 | 49 |
|  | noise $\sigma$ | 13 | 10 | 13 |
|  | **threshold** (background/LOD + 3 $\sigma$) | **184** | **59** | **88** |
| EMeRGe-Asia 2018 | upper tropospheric background | 145 | ~0 | ~0 |
|  | LOD | 22 | 15 | 25 |
|  | noise $\sigma$ | 18 | 15 | 11 |
|  | **threshold** (background/LOD + 3 $\sigma$) | **199** | **63** | **61** |

**Table 3: Definition of emission signatures that can be derived from measured enhancements of acetonitrile (ACN), benzene (BEN) and isoprene (ISO) above (1) or below (0) specified threshold levels (listed in Table 2). The emission signatures *aged BB*, *BB & BEN* and *AP* are defined only based on the combination of ACN and BEN. Other signatures are based on the combination of ACN, BEN and ISO.**
Note: Periods of instrumental background detection (see section 3.1.1) and when tracers change between 1 and 0 (or vice versa) are excluded and indicated with NA (last line).

| tracer enhancements | | | emission signatures | | | |
|---|---|---|---|---|---|---|
| ACN | BEN | ISO | I | II | abbreviation | description |
| *1* | *0* | *0* | *aged BB* | only BB | ACN | acetonitrile |
| *1* | *0* | *1* |  | only BB & BIO | BEN | benzene |
| *1* | *1* | *0* | *BB & BEN* | only BB & BEN | ISO | isoprene |
| *1* | *1* | *1* |  | BB & BEN & BIO |  |  |
| *0* | *1* | *0* | *AP* | only BEN | BB | biomass burning |
| *0* | *1* | *1* |  | only BEN & BIO | BIO | biogenic |
| 0 | 0 | 1 | only BIO |  | AP | anthropogenic |
| 0 | 0 | 0 | BG |  | BG | background |
| - | - | - | NA |  | NA | not assessable |

Hence, life-times of the selected VOCs do not only determine their atmospheric background levels but also define the time ranges during which certain emission signatures are unambiguously attributable to their sources. Considered together, we can roughly assess the stage of processing.

Based on the three selected VOCs, various emission signatures as described exemplarily above can be defined, which allows investigating and characterising source contributions during EMeRGe. Due to the PTR-MS duty cycle (consecutive integration of nine VOCs with ~6 s each), we interpolated the identification of emission signatures to the general measurement time resolution of one second. Inevitably, the interpolation can lead to gaps in the identification of signatures during instrumental background detection and when VMRs vary around the threshold (see supplement for detailed description). However, identified signatures can be used to filter other trace gas measurements of EMeRGe in order to analyse the chemical properties of air-masses of different origin.



### 3.2 Identification of source regions using back-trajectories and model analyses

To attribute sampled trace gas enhancements to certain source regions and to estimate their emission, we combine two tools:

**Tool 1:** Back-trajectories calculated with the FLEXTRA model (FLEXible TRAjectory model, Stohl et al., 1995, 2005; Stohl and Seibert, 1998), PBL heights from the ERA5 reanalysis (Hersbach et al., 2020) and

**Tool 2:** Mapping of CO emission rates based on the widely used inventory EDGAR (Emission Database for Global Atmospheric Research, https://data.jrc.ec.europa.eu/collection/edgar).

### 3.2.1 Tool 1: FLEXTRA back-trajectories and ERA5 PBL height

Using FLEXTRA, 10-days back-trajectories, with release steps of one minute, are calculated along the HALO flight tracks. The time-step along each trajectory is 10 minutes. Additionally, we use the ERA5 PBL height which is diagnosed as the height where the bulk Richardson number reaches the critical value of 0.25 (ECMWF, 2020). PBL height uncertainties can exceed 50% for shallow boundary layers (<1 km, e.g. at night). However, for deeper boundary layers the uncertainty is below 20%, e.g. at daytime (Seidel et al., 2012). We assume that the majority of emissions emanate from daytime activities in transportation

and traffic as well as energy use. Hence and due to the usually smaller daytime stability, PBL height uncertainties might mostly stay below 20%.

### 3.2.2 Tool 2: Mapping of CO emissions based on EDGAR emission inventory

The EDGAR emission inventory contains a variety of trace and greenhouse gas emission rates from different source categories on a worldwide grid (0.1° x 0.1°). For our analysis, we use the most recent version EDGARv6.1 Global Air Pollutant Emissions

(https://edgar.jrc.ec.europa.eu/dataset_ap61), which is based on a variety of regional emission inventories. From this data set, we selected anthropogenically emitted CO on a monthly basis for the year 2018 and considered a tropospheric lifetime of two months (Khalil and Rasmussen, 1990) to identify source regions contributing with recent and aged emissions (advected by long-range transport, LRT).

Figure 2a shows anthropogenic EDGAR CO emission rates for Europe in July. It indicates that emissions of the selected target

areas are high and overall regionally well confined. In contrast, CO emission rates for March in Asia show coherently high emissions in East and Northeast China (see reddish colours in Fig. 2b). The triangle Xian-Beijing-Shanghai is already described as a gigacity, consisting of megacity agglomerations (Kulmala et al., 2021) which challenges the identification of boundaries of distinct target megacities there.




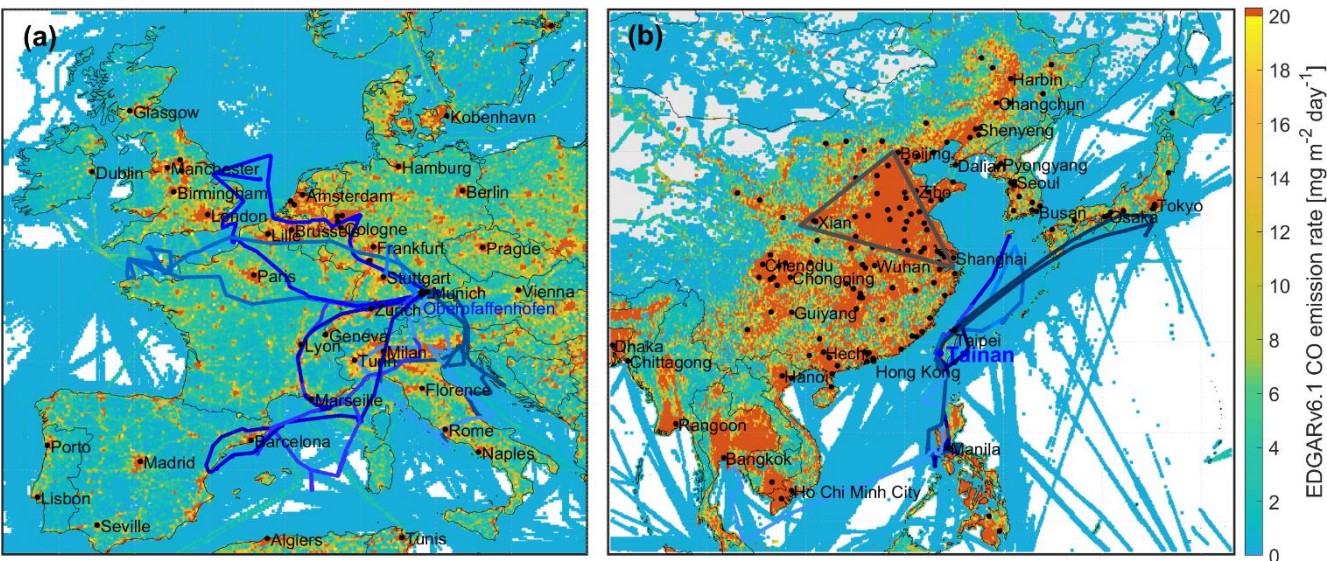

**Figure 2: CO emission rates from anthropogenic sources (EDGAR v6.1 Global Air Pollution Emission, year 2018, monthly means), (a) Europe: July (EMeRGe-Europe) (b) Asia: March (EMeRGe-Asia), and EMeRGe flight tracks (blue lines). Black dots mark cities with more than one million inhabitants. Grey triangle in (b) marks the megacity agglomeration between the cities Xian, Beijing and Shanghai. For clarity, emission rates below 0.01 mg m$^{-2}$ day$^{-1}$ are not shown.**

### 3.2.3. Modelled CO uptake from the PBL

To quantify the contribution of certain anthropogenic source regions during EMeRGe, we consider the anthropogenic CO emission rates provided by EDGAR and try to estimate the uptake of a certain amount of these CO emissions by back-trajectories passing by (see explanation in Fig. 3). By far, most emissions occur at or near the ground and are therefore emitted into the Earth's PBL. Thus, at first order, the CO uptake of a trajectory air-mass will be determined by the PBL residence time and the strength of emission rates there. While in the PBL, the horizontal advection by prevailing winds is still important. In compact regions with continuous emissions (or with one strong source and only slow horizontal advection) the volume mixing ratios in an air-mass are increased due to the accumulation of emissions. Other processes, like heterogeneous mixing with other air-masses and oxidation act as well and modify volume mixing ratios.


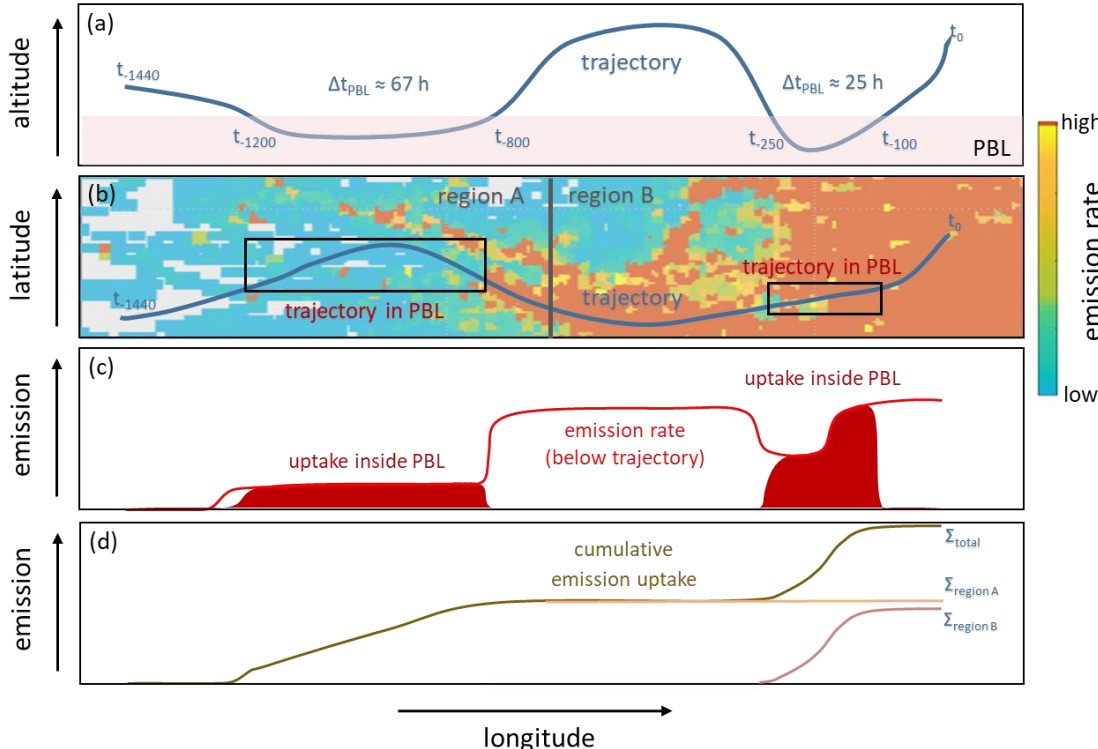


**Figure 3: Graphical representation (simplified example) of our Lagrangian CO uptake model based on PBL height and trajectory pathways, with longitude as x-axis. (a) Trajectory altitude. An exemplary trajectory (sampled by the aircraft at time $t_0$) enters the PBL for two different periods, between $t_{-1200}$ and $t_{-800}$, as well as between $t_{-250}$ and $t_{-100}$ ($t_{-100}$ means: 100 backward calculation steps = 1000 min = 16.7 h). During these periods, CO is taken up, that is, the trajectory is loaded with CO emitted on ground. (b) Geographic**
**map: CO emission rates from EDGAR (shaded) with a trajectory (blue line) and rectangles in which the trajectory immerses into the PBL. Exemplary, two example regions A and B are defined. (c) Time series: emission rate along the trajectory (red line) and CO uptake periods (filled red regions) along the trajectory. (d) Total and subtotal (region A and B) cumulative CO uptake (dark yellow line) along the trajectory.**

Based on this approach, we amalgamate the PBL residence time from FLEXTRA back-trajectories with the CO emission rates
(kg m$^{-2}$ s$^{-1}$) from the EDGAR inventory to estimate the potential contribution of certain emission regions to our observations. To derive an uptake for a back-trajectory inside the PBL at a certain time $t_x$, we calculate emissions (based on the emission rate of the corresponding grid box) for the area defined by the covered distance from trajectory point $t_x$ to $t_{x-1}$ multiplied by a "standard" width of 1 m, and for the fixed time-step of the trajectory from point $t_x$ to $t_{x-1}$ (10 minutes). These uptakes (given as mass) are calculated for all trajectory points inside the PBL and are summed up for all single 0.1° x 0.1° grid boxes. This
allows the identification of regions where emissions have a high potential to contribute to the probed air-masses during EMeRGe. Subsequently, summing up the amount of pollutants (here anthropogenic CO) taken up along trajectories in the PBL (at or near major source regions) prior to sampling should help to identify source regions that caused pollution sampled during EMeRGe and, in turn, should allow assessing the validity of our approach. Please note that such a straightforward Lagrangian





approach is only indicative, because it neglects the loss of CO by oxidation and dilution processes during transport and thus
the inferred CO uptake should be interpreted as a maximal CO emission contribution potential.

### 3.3.     AMTEX CO measurements

We use in situ CO measurements on-board HALO to qualitatively compare with the modelled CO emission uptakes. The
instrument is an AeroLaser 5002 vacuum UV resonance fluorescence spectrometer, installed on HALO, detecting the
resonance fluorescence in the fourth positive band of CO (Gerbig et al., 1996, 1999). Measurements are performed at 1 Hz
with a precision of 1.5 ppb. The total uncertainty is 1.5 ppb ± 2.4% (Gerbig et al., 1999).

### 4.     Results

First, we present the chemical characterisation of sampled air-masses based on our VOC measurements. Afterwards we use
FLEXTRA trajectories and the EDGAR CO inventory to identify source regions and to assess their contributions to the
modelled CO uptakes per flight and for the campaigns (EMeRGe-Europe, EMeRGe-Asia) in total. Finally, we link significant
CO uptakes of source regions to the VOC measurements in order to characterise their chemical fingerprints.

### 4.1.     VOC-based characterization of sampled air-masses

As introduced in section 3.1.2, the three VOCs, acetonitrile, benzene and isoprene, allow us to identify emission signatures in
sampled air-masses. Figure 4 illustrates the fractions of emission signatures observed during all 21 EMeRGe flights in Europe
and Asia (values are listed in the supplement, Table S2).
The most striking difference are the more frequent enhancements in acetonitrile and benzene during EMeRGe-Asia (18 % and
51 %, respectively) compared to EMeRGe-Europe (2.5 % and 22 %, respectively), see Fig 4a. In contrast, enhancements of
isoprene (of biogenic origin) were more often observed in EMeRGe-Europe (7.5 % versus 4 %). Here, the short lifetime of
isoprene (only some hours) might be the reason for the smaller percentages.  However, all fractions differ considerably from
flight to flight. Figure 4b, c show the partitioning of inferred emission signatures (defined in Table 3), with the following
overarching features.

**EMeRGe-Europe:** Around 21% of the flight time, polluted air has been sampled (EM-EU, Fig. 4b), with ~18 % of
anthropogenic, 2 % of BB & BEN and ~1 % of pure biogenic origin. Around 40 % of sampled air-masses comprise no VOC
tracer enhancement. The remaining 39 % cannot be attributed due to data gaps (see section 3.1.2).

In detail, benzene originates mainly from anthropogenic sources (4.5 % together with biogenic signals, ~ 10 % without),
relevant BB sources play a minor role. Nonetheless, the small fraction of BB & BEN, indicates either fresh BB or mixing of
BB and anthropogenic sources. Aged BB contributions are negligible. In most cases, biogenic isoprene signals are detected
together with anthropogenic benzene signals (only BEN & BIO, Fig. 4c), indicating a mixture of both with recent sources from
the PBL. A special feature is seen on flight EU-04, where the largest fraction besides BG are pure biogenic signatures (~8 %),





and aged BB (1.3 %). During this flight no MPC was targeted, but a direct flight-to-flight comparison of HALO with the
British FAAM was conducted (Schumann, 2021).

**Figure 4: Fractions [%] of (a)** enhancements over thresholds observed in acetonitrile, benzene and isoprene, **(b)** emission signatures
I (*aged BB*, *BB & BEN* and *AP* inferred from observed enhancements only in acetonitrile and benzene; *only BIO* and *BG* inferred
with additional isoprene) and **(c)** emission signatures II (inferred from observed enhancements in acetonitrile, benzene and isoprene).
**Top: during EMeRGe-Europe, bottom: EMeRGe-Asia.**
AP – anthropogenic signatures, BB – biomass burning signatures, BEN – benzene enhancements, BIO – fresh biogenic signatures,
BG – background, NA – not assessable during instrumental background detection and threshold transitions (due to PTR-MS
measurement resolution of one minute per tracer). EU-05 is not available due to instrument failure. The summary of EMeRGe-Asia
excludes the non-Asian transfer flights AS-01 and AS-16.





**EMeRGe-Asia:** Around 50% of the flight time, polluted air has been sampled (EM-AS, Fig. 4b), with ~32 % of anthropogenic, ~14 % of BB & BEN and ~2 % aged BB origin. Around 15 % of sampled air-masses comprise no VOC tracer enhancement. The remaining ~37 % cannot be attributed.

In detail, benzene enhancements originate on average to two-thirds from anthropogenic signals and are detected to one-third together with BB (BB & BEN). However, the variation is large from flight to flight. Several flights (e.g. AS-07, -08 and -09) show predominantly AP signatures. When BB influence is strong (e.g. AS-05, -06, -10 and -14), BB & BEN predominate. We assume that in these cases, not only fresh BB contributes to BB & BEN but also anthropogenic sources, because the anthropogenic signature AP is in these flights significantly smaller compared to other flights with less BB. During several EMeRGe-Asia flights (e.g. AS-03, -06, -12 and -14), we detected also aged BB signals showing the long-range transport of BB events.

In comparison, EMeRGe-Europe possesses only minor BB contributions identified mostly on flights in Southern Europe, whereas measurements during EMeRGe-Asia are more regularly and stronger influenced by BB events due to the active BB season in Asia. The fraction of benzene enhancements (percentage of measurements over benzene threshold) is in EMeRGe-Asia more than double that of EMeRGe-EU and primarily of anthropogenic origin. The fraction of measurements with no tracer signals (BG) is considerably higher in EMeRGe-Europe than EMeRGe-Asia, which means that more 'clean' background air was probed during EMeRGe-Europe.

With such a use of tracer combinations, 50 - 70 % of the EMeRGe observations can be attributed to certain emission signatures. As shown by the white bars (representing not assessable (NA) measurements) in Fig. 4b, the attribution capacity differs from flight to flight and depends on the availability of tracer combinations (either two or three tracers and on the interpolation mentioned in section 3.1.2).

Overall, the consideration of different chemical tracers enables a sophisticated characterisation of potential emission sources from airborne composition measurements.

## 4.2.    Source regions

### 4.2.1.    Trajectory- and inventory-based identification of source regions

In order to identify air-mass origins of observed pollution events, we use the CO uptake approach outlined in section 3.2.3 (combining the EDGAR emission inventory with FLEXTRA back-trajectories) to assess the potential emission contribution of certain source regions to the probed air-masses.

Figure 5 and 6 show (a) the EDGAR CO emission rates of anthropogenic sources for the EMeRGe campaign periods, (b) the cumulative residence time of back-trajectories inside the PBL for every EDGAR grid box, and (c) the modelled CO uptake from certain source regions.

**Figure 5: (a) EDGARv6.1 CO emission rates from anthropogenic sources (July 2018), for clarity, not colour-coded below 0.01 mg m$^{-2}$ day$^{-1}$, (b) PBL residence time of air parcels cumulated for a grid resolution of 0.1 ° x 0.1 °, (c) modelled CO emission uptake from the PBL interpreted as emission contribution potential during EMeRGe-Europe. Boxes in (c) mark selected study areas (summarised in Table, left). Black dots mark cities with more than one million inhabitants, the blue dot denotes the home base of HALO (airport Oberpfaffenhofen near Munich, Germany) during EMeRGe-Europe.**



**Figure 6: (a)** EDGARv6.1 CO emission rates from anthropogenic sources (March 2018), for clarity, not colour-coded below 0.01 mg m⁻² day⁻¹, **(b)** PBL residence time of air parcels (for flights AS-03 to AS-10, AS-12, AS-13) cumulated for a grid resolution of 0.1 ° x 0.1 °, **(c)** calculated CO emission uptake from PBL interpreted as emission contribution potential during EMeRGe-Asia. Boxes in c) mark selected study areas (summarised in Table 4, right-hand side). Black dots mark cities with more than one million inhabitants, the blue dot denotes the home base of HALO (airport Tainan, Taiwan) during EMeRGe-Asia.





**EMeRGe-Europe:** In general, strong anthropogenic emission hot spots are located in Central Europe and the Eastern United States (Fig 5a). We found most frequent PBL contact of air-masses probed during EMeRGe-Europe (Fig. 5b) near target regions of Spain/Southern France, Northern Italy and South Great Britain. Due to the west wind drift, PBL contact of air parcels also occurred over the Atlantic Ocean as well as over the USA and Canada, indicating the influence of long-range transport (LRT).

We identified the largest contribution potentials of anthropogenic CO emissions (Fig 5c) in Southern Great Britain, including London, Northern Italy, the Eastern USA, and some hot spots like the Ruhr area, Madrid and parts of Southern France. Besides the seven target MPCs in Europe, we also selected 17 additional geographical areas with enhanced contribution potentials, with smaller (better resolved) emission grids around the target MPCs. All not-categorised regions are combined as unspecified areas. Table 4 (left) gives a summary of all selected source regions.

**EMeRGe-Asia:** More regions have likely contributed to the observed composition during EMeRGe-Asia which occurred in the outflow of the entire Eurasian continent with some LRT even originating from Europe. However, the coherently highest CO emission rates ($\geq 20$ mg m$^{-2}$ day$^{-1}$) emanate from India and East China (Fig. 6a).

We found PBL contact of probed air-masses (Fig. 6b) over Southeast Asia, China, India as well as over Southern Europe, Northern Africa and Western Asia. However, probed air-masses had the most prolonged PBL contact over the East China Sea
close to the sampling region.

We identified the largest contribution potentials (Fig. 6c) in China, especially in the gigacity triangle Xian-Beijing-Shanghai and in Northeastern and Southern parts of China. Furthermore, Taiwan, Japan, Thailand, India and parts of Vietnam show enhanced contribution potentials. We also found widespread uptakes in South and Southeast Europe, Northern Africa and Western Asia, indicating that measurements during EMeRGe-Asia are partly influenced by LRT as well. Although the East
China Sea has the most prolonged PBL contact, contribution potentials are small, most likely due to much smaller ship emissions compared to emissions onshore. The selected source regions are summarised in Table 4 (right).

**Table 4: Source regions and MPCs (*italic*) selected for EMeRGe-Europe and -Asia. Coordinates are listed in the supplement (Table S3).**

| | | EMeRGe-Europe | | | EMeRGe-Asia |
|---|---|---|---|---|---|
| no. | abbr. | source region/*MPC* | no. | abbr. | source region/*MPC* |
| 1 | CAN | Canada | 1 | EUA | Europe/Northern Africa |
| 2 | USA | United States of America | 2 | WAS | Western Asia |
| 3 | NAT | North Atlantic Ocean | 3 | WCH | Western China |
| 4 | IRE | Ireland | 4 | IND | India |
| 5 | NGB | Northern Great Britain | 5 | CCH | Central China |
| 6 | SGB | Southern Great Britain | 6 | SCH | Southern China |
| 7 | BNR | Belgium, Netherlands and Ruhr area (BeNeRu) | 7 | SEA | Southeast Asia |
| 8 | NFR | Northern France | 8 | MOR | Mongolia/Southern Russia |
| 9 | SFR | Southern France | 9 | ECH | East China |
| 10 | IBE | Iberian Peninsula | 10 | TAW | Taiwan |
| 11 | NGE | Northern Germany | 11 | NPH | Northern Philippines |
| 12 | SGE | Southern Germany | 12 | NEC | Northeastern China |
| 13 | NIT | Northern Italy | 13 | KOR | Korea |
| 14 | SIT | Southern Italy | 14 | JAP | Japan |





| 15 | NAF | Northern Africa | | 15 | ECS | East China Sea |
|---|---|---|---|---|---|---|
| 16 | NEU | Northern Europe | | | UA | Unspecified areas |
| 17 | EEU | Eastern Europe | | M1 | XBS | Δ Xian-Beijing-Shanghai (incl. in ECH) |
| | UA | Unspecified areas | | M2 | BEI | Beijing (incl.in XBS) |
| M1 | LON | London (incl.in SGB) | | M3 | YAN | Yangtze River Delta (incl.in XBS) |
| M2 | PAR | Paris (incl.in NFR) | | M4 | PEA | Pearl River Delta (incl.in SEC) |
| M3 | MAD | Madrid (incl.in IBE) | | M5 | TOK | Tokyo (incl.in JAP) |
| M4 | BAR | Barcelona (incl.in IBE) | | M6 | OSA | Osaka (incl.in JAP) |
| M5 | POV | Po Valley/ Milano (incl.in NIT) | | M7 | BAN | Bangkok (incl.in SEA) |
| M6 | ROM | Rome (incl.in SIT) | | M8 | MAN | Manila (incl.in NPH) |
| M7 | MUN | Munich (incl.in SGE) | | M9 | TAI | Taipei (incl.in TAW) |

### 4.2.2. Contribution potentials of source regions

After we identified potential source regions, we analyse in this section to which degree their emissions have contributed to EMeRGe flights and on which days prior to the HALO measurements the uptakes occurred. Figure 7 and 8 summarise relative (colour-coded) and absolute modelled CO contribution potentials during EMeRGe-Europe (seven flights, 17 source regions and seven target MPCs) and EMeRGe-Asia (12 flights, 15 source regions and nine MPCs), respectively, as listed in Table 4. Please note that absolute CO contribution potentials in Fig. 7 and 8 should be interpreted with care since the magnitude depends on the assumptions and weighting of the emission uptake. Nonetheless, a relative representation is useful to assess individual source region contributions. Note furthermore, that they reflect only the sampling (in space and time) of the atmosphere by the flights and the emission rates/climatology of the specific month.

**EMeRGe-Europe.** Most of the flights have successfully probed pollution from target regions (Fig. 7a and c). However, some target regions apparently had only small contributions, whereas some non-target regions contributed considerably more. In general, EU-08 with the targets London and BeNeRuhr has the largest modelled CO emission uptake and EU-04, targeting Southern Germany near the Alps (no MPC target and shortest flight) the smallest.

Figure 7b indicates further that around 50 % of the modelled total CO uptake sum $\sum_{EM-EU}$ has been emitted in the last three days prior to the measurement, that is, from South Great Britain, BeNeRuhr, Southern/Northern France, the Iberian Peninsula, Southern Germany and Northern Italy. Emission uptakes of 4 to 8 days prior to the HALO measurements originate mainly from BeNeRuhr, Northern France, the Iberian Peninsula and Eastern Europe. Earlier uptakes (8 to 10 days prior to HALO measurements) have arisen mainly from the USA and Canada, but with overall small cumulative contribution potentials. In total, fresh emissions from the last 72 hours as well as transported emissions contributed to probed air-masses during EMeRGe-Europe. In total, Belgium-Netherlands-Ruhr (14 %), South Great Britain (13 %) and Southern France (12 %) had the largest CO emission contribution potentials during EMeRGe-Europe accounting for around 40 % of the uptake sum $\sum_{EM-EU}$. All target MPCs together contributed up-to 16.5 %.

In the following, we analyse the contribution potentials of specific target regions and MPCs in more detail. Flights EU-03 and -06 investigated the outflow of the Po Valley and Rome. The Po Valley contributed with 10 and 15 % (Fig. 7c), and dominates the total uptake over Northern Italy (17 and 23 %, respectively, Fig. 7a). The outflow of Rome and Southern Italy was only occasionally probed, with contributions of 4 % and 3 %. The largest contributors of non-target areas during EU-03 are the Iberian Peninsula (22 %) and Northern Africa (17 %). During EU-06, emissions of Eastern Europe dominated with 32 %.



**Figure 7: Absolute and relative CO contribution potentials of 17 source regions and seven MPCs (listed in Table 4) during EMeRGe-Europe. Absolute (plain) values give the sum of contribution potentials for single flights (*abs.* ∑$_{EU-XX}$), uptake days (*abs.* ∑$_{day}$) as well as for source regions and MPCs (*abs.* ∑$_{region}$). For EMeRGe-Europe, the total uptake sum is 42.0 kg (*abs.* ∑$_{EM-EU}$), for the selected MPCs 6.9 kg (*abs.* ∑$_{MPC}$). Small deviations between absolute value sums and the total uptake sum are caused by rounding. Relative contribution potentials (colour-coded) of individual source regions are separated into contributing emission uptakes per flight (a, c), normalised to the absolute uptake sum per flight (*region*$_{EU-XX}$/*abs.* ∑$_{EU-XX}$), as well as separated into contributing emissions for the 10 particular uptake days prior to sampling with HALO (b, d), normalised to the absolute uptake sum per uptake day (*region*$_{day}$ /*abs.* ∑$_{day}$). Relative uptake sums *rel.* ∑$_x$ are normalised to the total uptake sum (*rel.* ∑$_x$ = *abs.* ∑$_x$/∑$_{EM-EU}$, with x=EU-XX, *day* and *region*). Target regions are given in the column label after flight labels. Regions are sorted by location from West (top) to East (bottom). Unspecified includes all areas outside the selected regions.**

During flight EU-04, HALO conducted an inter-comparison flight with the British FAAM research aircraft (Schumann, 2021) in Southern Germany (Allgäu), as mentioned before. We identified contributions from Southern Germany (38 %, primarily by Munich with 35.5 %) and Southern Great Britain (13 %), but also LRT contributions from USA (19 %) and Canada (11 %). However, the modelled total CO uptake for flight EU-04 is small compared to other flights.

Flights EU-05 and -08 investigated the London area. However, the actual emission uptake modelled was small (~5-8 %) for both flights. During EU-05, emission uptakes mostly arose from South Great Britain (73 %). Around 13 % originated from the





USA by LRT. Flight EU-08 investigated the Belgium-Netherlands-Ruhr area as well, which contributed up-to 44 %, whereas

South Great Britain showed only a contribution of 13.5 %.

The flights EU-07 and -09 investigated the outflow of Madrid and Barcelona, where Madrid contributed during EU-09 with 9 %. During EU-07 contributions from Madrid and Barcelona are negligible. Emissions of the surrounding Iberian Peninsula contribute to 10 and 23 %, respectively. Non-target contributions originate from Northern and Southern France, where Paris dominates emissions of Northern France during EU-09. Additionally, LRT emissions from the USA contributes with 8 and

11.5 % respectively.

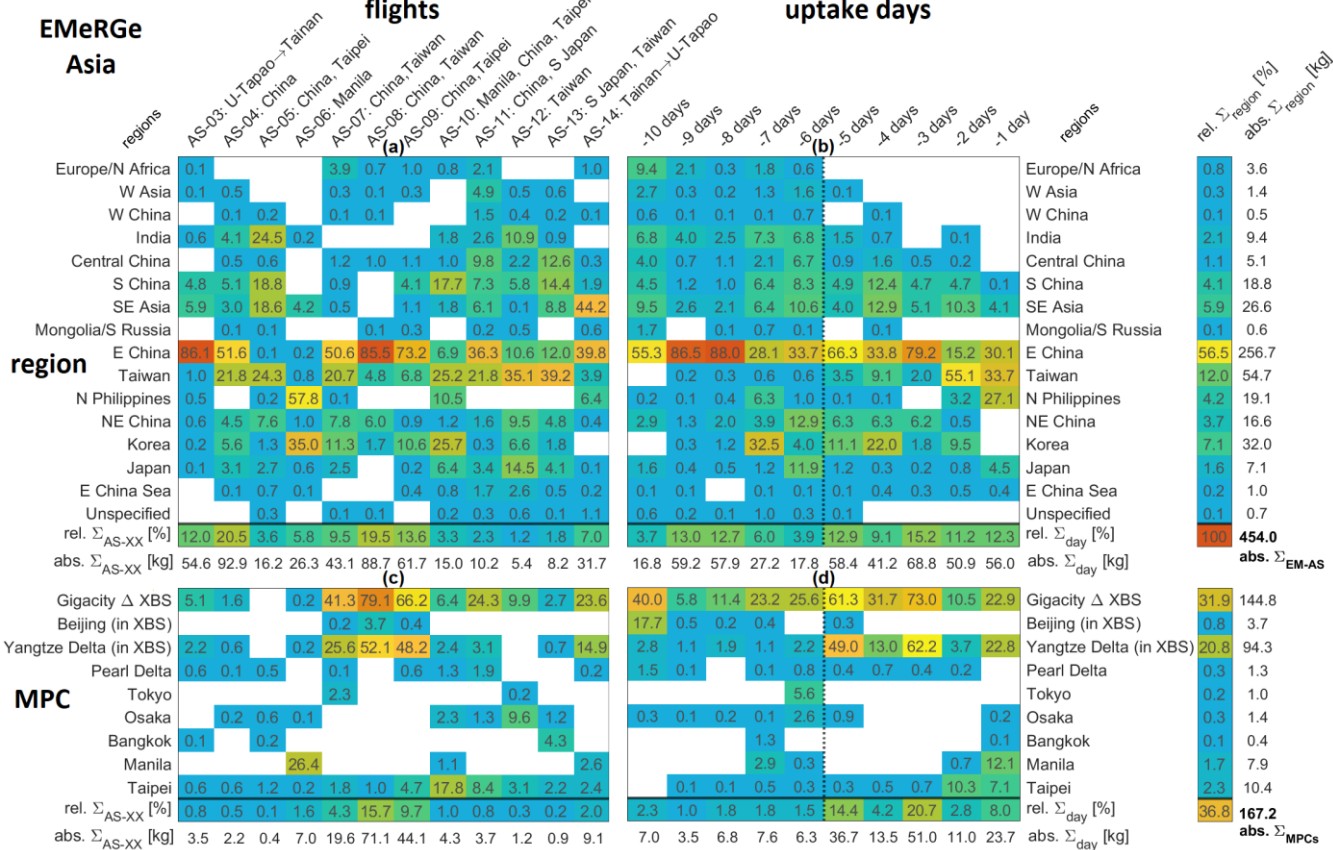

**Figure 8: Absolute and relative CO contribution potentials of 15 source regions and nine MPCs (listed in Table 4) during EMeRGe-Asia. Absolute (plain) values give the sum of contribution potentials for single flights (_abs. $\sum_{AS-XX}$_), uptake days (_abs. $\sum_{day}$_) as well as for source regions and MPCs (_abs. $\sum_{region}$_). For EMeRGe-Asia, the total uptake sum is 454.0 kg (_abs. $\sum_{EM-AS}$_), for the selected MPCs**

**167.2 kg (_abs. $\sum_{MPC}$_). Small deviations between absolute value sums and the total uptake sum are caused by rounding. Relative contribution potentials (colour-coded) of individual source regions are separated into contributing emission uptakes per flight (a, c), normalised to the absolute uptake sum per flight (_region$_{AS-XX}$/abs. $\sum_{AS-XX}$_), as well as separated into contributing emissions for the 10 particular uptake days prior to sampling with HALO (b, d), normalised to the absolute uptake sum per uptake day (_region$_{day}$ /abs. $\sum_{day}$_). Relative uptake sums _rel. $\sum_x$_ are normalised to the total uptake sum (_rel. $\sum_x$ = abs. $\sum_x/\sum_{EM-AS}$, with x=AS-XX, day and region_).**

**Target regions are given in the column label after flight labels. Regions are sorted by location from West (top) to East (bottom). Unspecified includes all areas outside the selected regions. XBS – Xian-Beijing-Shanghai.**





**EMeRGe-Asia:** Compared to EMeRGe-Europe, modelled emission uptakes are much larger and vary more from flight to flight (Fig. 8a and c). AS-12 shows the smallest uptake (targeting Taiwan) comparable with the EMeRGe-Europe flights. AS-04 has the largest uptake (targeting Chinese outflows), 17 times larger than that of AS-12. Furthermore, in relation to the completed flight hours (with comparable flight characteristics, see Fig. 1), the uptakes during EMeRGe-Asia are ~6 times larger than during EMeRGe-Europe, representing the overall large emission sources probed during EMeRGe-Asia.

As indicated in Fig. 8b and d, large contributions are from uptakes on the first five days prior to probing (61 % of $\sum_{EM-AS}$). As expected, air-masses from Taiwan (Taipei), East China and Northern Philippines (Manila) comprise mostly signatures of fresh emissions due to probing in their proximity. East China has the largest contribution for nearly all uptake days, showing a constant large emission outflow. The small contributions of Europe/Northern Africa, Western Asia and India are mostly taken up six and more days prior to probing with HALO, indicating LRT.

As one of its main targets, EMeRGe-Asia successfully probed the outflow of Mainland China (contribution of ~66 %), which is unambiguously represented by the large contribution from the source region East China with ~57 %. Emission uptakes of other source regions thus appear much smaller, on average, e.g. Taiwan (12%), Korea (7 %), Southeast Asia (6 %), Northern Philippines, South and Northeast China (each ~4 %). However, much stronger contributions have been sampled during individual flights.

A further aim was the probing of recent and transported outflows of Asian MPCs. Uptakes over the Yangtze Delta show an overall contribution of ~21 %, Taipei and Manila of 2 %. The MPC agglomeration in between the triangle Xian-Beijing-Shanghai (including the MPCs Beijing and Yangtze Delta) shows the largest emission contribution during EMeRGe-Asia with ~32 % and is mostly dominated by uptakes of Yangtze Delta in the first five days prior to probing. The MPC emissions are taken up mainly during flights AS-07 to -09 (Fig. 8c). Uptakes from Taipei contributed in small fractions during most of the flights and mainly as recent emissions (Fig. 8d) due to the close location to the flight base Tainan. Emission uptakes of the other listed MPCs are negligible. All uptakes of MPC emissions contribute to ~37 %.

In summary, the measurement approach of EMeRGe covered large regions during each flight and "jumping" from MPC to MPC resulted in a large spectrum of sampled air-masses. During each flight different source signatures have been observed, that is, from MPCs (the initial target areas), but also from surrounding regions as well as from emissions transported over longer distances. Likewise, the degree of chemical processing and mixing varied. A more detailed inspection of MPC outflows is difficult and would require more close proximity flights. However, due to flight restrictions close to MPC airspaces this is not easily realised. In contrast, the regional-scale measurement approach nicely enabled to cover the full range and thus fate of air-masses, from fresh emission signatures close to MPCs, to chemically processed and already mixed air-masses to background air. The results of EMeRGe thus allow us to provide some general recommendations on the observation strategy of future aircraft campaigns (see section 5, conclusions and summary).





### 4.2.3. Linking and partitioning modelled source region emissions to observations

To analyse source region specific fingerprints, we link the trajectory- and inventory-based, anthropogenic CO uptakes with
the EMeRGe observations. First, we outline our systematic approach (comprising the following three steps) and secondly, we
will present the analyses. The aim is to distinguish observation periods that are significantly influenced by emissions of our
EMeRGe target regions from other source region emissions.

**Step 1:** We sum up the modelled source region-specific CO uptakes along the trajectory pathway (their release points have
one-minute time resolution along the HALO flight track, as explained in section 3.2.3 and Fig. 3) and assign and distribute
these uptake sums equally to 60 seconds centred at the trajectory release point (assuming that the trajectory is representative
for the minute). This is done for every trajectory, to cover the complete measurement time resolution (one second) of HALO.
Figure 9 illustrates this first step for the flight EU-06, linking the modelled CO uptake sums with observations.

**Step 2**: Next, we analyse the contribution of the individual source regions to the CO uptakes per trajectory. To consider only
significant CO uptakes, we omit small ones, which in sum add up to 5 % of the total CO uptake sum of the respective campaign
part, that is, ~2.6 g individual CO uptake per region for EMeRGe-Europe and ~20.6 g CO for EMeRGe-Asia. Afterwards, the
source region partitioning of every trajectory uptake is examined. Hence, a CO uptake sum can comprise emissions of a single
or of a few source regions. In the following, we will refer to CO uptake sums consisting of single source region emissions as
"non-mixed" and CO uptake sums consisting emissions from various regions as "mixed".

**Step 3:** Summarising all individual contribution patterns derived in step 2, again assigned to the corresponding observations
(in time and space) as mentioned in step 1, results in frequency distributions of source regions which contribute with non-
mixed or within mixed emissions to the EMeRGe observations Table 5 lists these contributing source regions and MPCs. We
additionally provide an equivalent list in the supplement (Table S4), where we also consider the small CO uptakes, omitted in
step 2. This shows that even more source regions contribute to the "mixed emissions". These additional regions are often large
and have small emission rates and thus small uptakes, like the North Atlantic during EMeRGe-Europe or the East China Sea
during EMeRGe-Asia.

As mentioned above, Fig. 9 illustrates step 1 for the flight EU-06. Note that the modelled CO uptakes are not directly
comparable to the measured CO, because chemical decay and dilution processes are missing in the simple uptake approach.
Therefore, they are not able to reflect temporal variations or amplitudes comprehensively. However, large and small CO uptake
sums (Fig. 9b) show a reasonable coincidence to observed CO enhancements (above a background of 80-90 ppbV).
Contributions from different source regions (Fig. 9c) also depend on sampling altitudes. At altitudes below 1500 m, nearby
emissions of Eastern Europe (pink), Po Valley (dark purple), Northern Italy (light purple), Southern Italy (dark green) and
Rome (sea green) dominate, accompanied by enhanced benzene concentrations from recent anthropogenic emissions.

However and as expected, BB signatures also emerge in air-masses not linked to CO uptakes (e.g. around 13:00 UTC), which
indicate local events not covered by the emission inventory (identified from MODIS (https://firms.modaps.eosdis.nasa.gov/)
as a local fire near Rome), as well as in the air-mass mixture of Rome, Northern Italy and Eastern Europe between 14:45 and

["





Atmospheric Chemistry and Physics
EGU
Discussions

15:30 UTC. Air-masses at higher altitudes above 2000 m originate mainly from the Iberian Peninsula (reddish colours) and Northern Africa (yellow), showing very small CO uptakes and correspondingly low measured CO concentrations between 70 and 90 ppbV.

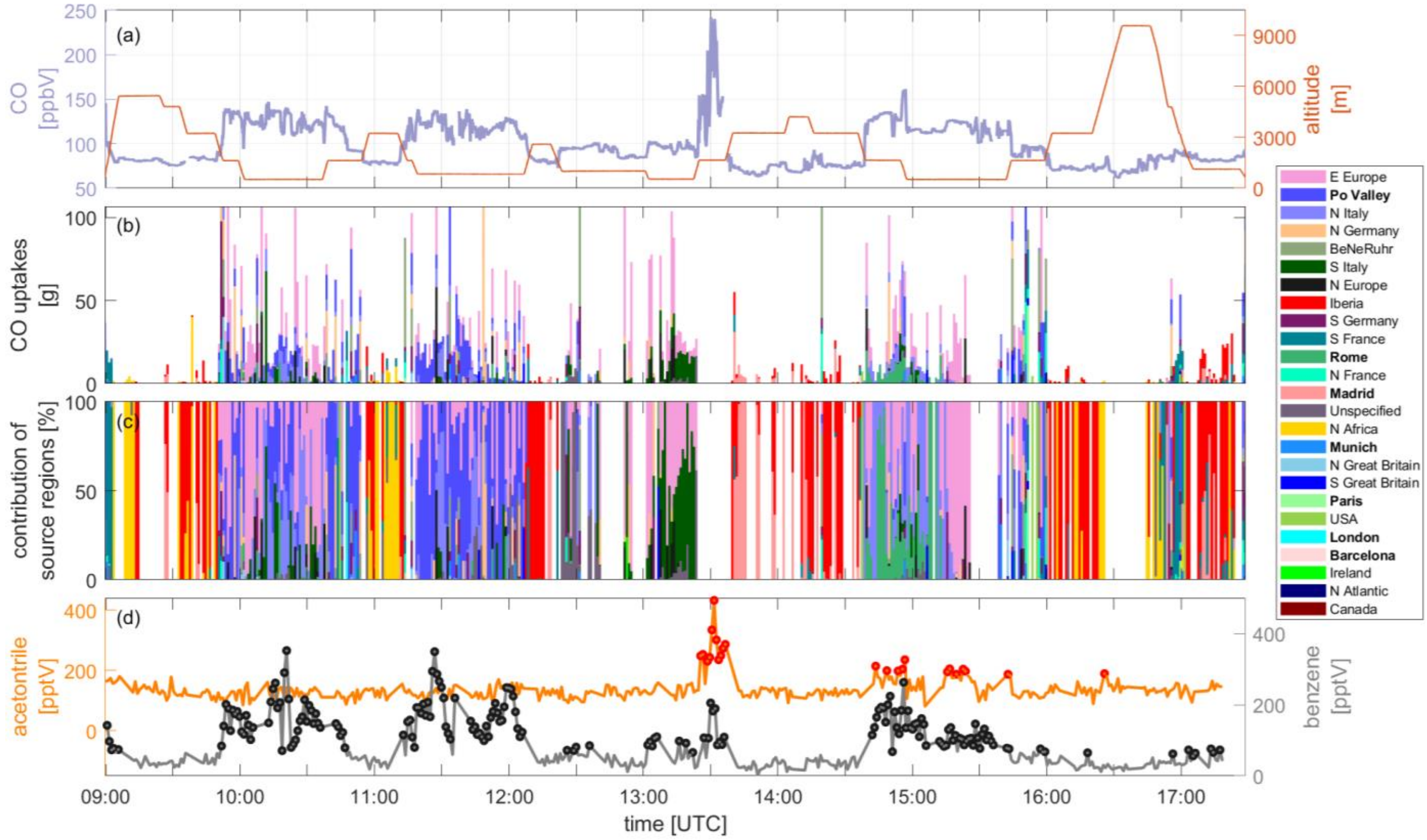

**Figure 9: Time series of modelled source region specific CO uptakes and measured CO, acetonitrile and benzene for EMeRGe-Europe flight EU-06, 20th July 2017, targeting Italian outflows. (a) CO measurements and flight altitude, (b) modelled CO uptake sums along 10-days back-trajectories colour-coded by originating source regions (sorted by total flight uptake sums – top to bottom, MPCs in bold), (c) relative contribution of source regions on CO uptake sums and (d) acetonitrile and benzene measurements with colour-coded tracer enhancements over defined threshold (red and black dots, respectively).**

Overall, Table 5 shows that during EMeRGe-Europe ~46 % of significant CO uptakes are linked to the observation period and 26 % during EMeRGe-Asia. The reason behind the smaller fraction during EMeRGe-Asia might be the larger absolute uptake value in step 2 (20.6 g CO uptake versus 2.6 g) for considering significant uptakes.

**EMeRGe-Europe:** Table 5 (left) indicates that the major temporal contributions are from Southern France, Iberian Peninsula, USA, Madrid and Southern Great Britain, that is, are from LRT within the west wind drift zone as well as from south- and north-western target regions in Europe. Often, these outflows dispersed during transport and were also sampled as "mixed emissions". Besides Madrid, also the MPC London, Munich (flight base), Po Valley and Paris contribute as significant emission sources. Emissions of London are mostly identified in conjunction with emissions from Southern Great Britain and Ireland (SGB-LON, IRE-SGB-LON), since HALO sampled predominately over the English Channel where emissions of these

regions have already mixed. The temporal contribution of Po Valley is rather small, although probed during two flights, similar
to Northern Italy, most likely indicating more confined air-masses due to probing in closer proximity. The Paris outflow is
most frequently identified in conjunction with emissions from Munich, probed close to the flight base (not shown), indicating
advection of the Paris outflow to Southern Germany. Altogether, we can obtain individual chemical fingerprints of the MPCs
Madrid, Munich, Po Valley and London.

**Table 5: Trajectory-based emission contributions from different source regions (left from EMeRGe-Europe, right EMeRGe-Asia), listed according to temporal frequency and magnitude (contribution in %, flight time in min, uptake sum in kg). Mixtures (of significant uptakes) from different sources regions are indicated by hyphens, e.g. "IRE-SGB". MPCs are highlighted with bold letters. Small uptakes, in sum 5 % of the total modelled uptake sum of the respective campaign part and contributions of less than 10 minutes flight time are omitted. The upper part indicates the overview (sums) of both campaigns. For full names of source regions see Table 4.**

| | **EMeRGe-Europe** | | | | **EMeRGe-Asia** | | |
|---|---|---|---|---|---|---|---|
| observations linked to | Contribution to total flight time [%] | linked flight time [min] | uptake sum [kg] | observations linked to | Contribution to total flight time [%] | linked flight time [min] | uptake sum [kg] |
| no up-take | 35.4 | 1103 | 0 | no up-take | 39.9 | 2248 | 0 |
| small up-takes filtered | 19.0 | 593 | 2.1 | small up-takes filtered | 34.4 | 1939 | 22.7 |
| residual uptakes | 45.6 (**19.8**) | 1424 (**616**) | 39.9 (**21.4**) | residual uptakes | 25.8 (**9.8**) | 1453 (**552**) | 431.3 (**268.0**) |
| *non-mixed (**MPCs**)* | *20.6 (**5.0**)* | *645 (**153**)* | *9.0 (**1.2**)* | *non-mixed (**MPCs**)* | *17.0 (**3.9**)* | *956 (**216**)* | *108.4 ( **21.6**)* |
| *mixture (**MPCs** involved)* | *25.0 (**14.8**)* | *779 (**463**)* | *30.9 (**20.2**)* | *mixture (**MPCs** involved)* | *8.8 (**5.9**)* | *497 (**336**)* | *322.9 (**246.4**)* |
| EM-EU (7 flights) | 100 | 3120 | 42.0 | EM-AS (12 flights) | 100 | 5640 | 454 |

| source region(s)/ **MPCs** | Contribution to total flight time [%] | linked flight time [min] | uptake sum [kg] | source region(s)/ **MPCs** | Contribution to total flight time [%] | linked flight time [min] | uptake sum [kg] |
|---|---|---|---|---|---|---|---|
| SFR | 3.3 | 102 | 2.4 | SEA | 3.1 | 177 | 17.9 |
| USA | 2.8 | 89.0 | 1.2 | **XBS** | 2.1 | 116 | 16 |
| IBE | 2.8 | 89.0 | 0.8 | TAW | 1.8 | 104 | 22.9 |
| **MAD** | 2.4 | 76.0 | 0.4 | **TAI** | 1.5 | 85 | 4.4 |
| SGB | 1.7 | 53.0 | 0.9 | IND | 1.4 | 78 | 5.2 |
| IRE-SGB | 1.4 | 43.0 | 1.1 | SCH | 1.3 | 73 | 7.5 |
| SGB-**LON** | 1.2 | 38.0 | 1.4 | NPH-**MAN** | 0.9 | 52 | 15.2 |
| **MUN** | 1.0 | 31.0 | 0.3 | JAP | 0.8 | 48 | 2.9 |
| NEU | 0.9 | 28.0 | 0.4 | CCH | 0.8 | 47 | 2.6 |
| CAN | 0.8 | 25.0 | 0.2 | NPH | 0.8 | 44 | 1.9 |
| IBE-**MAD** | 0.7 | 22.0 | 0.3 | ECH-**XBS** | 0.7 | 39 | 10.2 |
| NAF | 0.7 | 22.0 | 0.3 | EUA | 0.6 | 36 | 1.8 |
| BNR | 0.7 | 21.0 | 1.1 | ECH | 0.6 | 34 | 4.3 |
| USA-SFR | 0.6 | 20.0 | 0.3 | KOR | 0.6 | 34 | 9.2 |
| BNR-NEU | 0.6 | 20.0 | 1.8 | TAW-**TAI** | 0.5 | 29 | 6.1 |
| SFR-IBE | 0.6 | 20.0 | 0.3 | **YAN** | 0.5 | 27 | 7.2 |
| NFR | 0.6 | 19.0 | 0.1 | SCH-SEA | 0.5 | 26 | 3.6 |
| **POV** | 0.6 | 18.0 | 0.2 | NEC-KOR | 0.4 | 25 | 8.9 |
| EEU | 0.5 | 17.0 | 0.3 | IND-SCH | 0.4 | 24 | 5.4 |
| IRE-SGB-**LON** | 0.5 | 16.0 | 0.6 | NEC | 0.4 | 24 | 2.6 |
| NIT | 0.5 | 16.0 | 0.1 | NEC-KOR-**YAN-XBS** | 0.2 | 14 | 5.4 |
| SIT-EEU | 0.4 | 14.0 | 0.4 | **YAN-XBS** | 0.2 | 14 | 2.9 |
| SFR-NIT | 0.4 | 13.0 | 0.4 | WAS | 0.2 | 12 | 0.8 |
| IBE-NAF | 0.4 | 13.0 | 0.3 | OSA | 0.2 | 12 | 0.9 |
| **PAR-MUN** | 0.4 | 13.0 | 0.5 | ECH-**YAN** | 0.2 | 11 | 3.7 |
| IBE-UA | 0.4 | 12.0 | 0.2 | | | | |
| **LON** | 0.4 | 11.0 | 0.1 | | | | |
| NIT-**POV** | 0.3 | 10.0 | 0.2 | | | | |
| UA | 0.3 | 10.0 | 0.1 | | | | |
| ∑ | 27.9 | 881 | 16.7 | ∑ | 20.7 | 1185 | 169.5 |





**EMeRGe-Asia:** The emissions of Southeast Asia, the gigacity Xian-Beijing-Shanghai, Taiwan, Taipei, India and South China contributed most to the sampled pollution events (Table 5, right), showing extensive probing of local emissions (Taiwan and Taipei), of short-range (Chinese outflow) and long-range transport (Southeast Asia and India). For EMeRGe-Asia, we can obtain fingerprints of individual MPC air-masses of Xian-Beijing-Shanghai, Taipei, Yangtze Delta, and Osaka.

The summary in the header of Table 5 shows that our trajectory-based source identification did not show CO uptake (in the last 10 days) for 35 % (EMeRGe-Europe) and 40 % (EMeRGe-Asia) of the flight time. However, these air-masses can also comprise emission signals, either of aged pollution or from sources not covered by the inventory. Moreover, in 19 % and 34 %, respectively, small/negligible contributions are inferred and not considered. Thus, the remaining flight time of 46 % (EMeRGe-Europe) and 26 % (EMeRGe-Asia) was characterized as polluted and attributed to certain source regions as outlined
below.

The temporal contribution of emission mixtures linked to observations is half as small during EMeRGe-Asia (11 %) compared to EMeRGe-Europe (25 %). However, uptake totals of mixtures relative to the campaigns total uptakes are comparable (~ 70 %), indicating that during EMeRGe-Asia especially emission mixtures of multiple source regions are involved in more compact but strongly polluted air-masses. MPC linkages are likewise comparable. Together, the linkage of significant MPC
emissions involved in uptakes is ~20 % during EMeRGe-Europe and ~10 % during EMeRGe-Asia. These numbers again confirm that EMeRGe probed emissions of distinct regions, emission mixtures of several regions, but also air-masses with no recent contact with emission sources in the PBL. However, mixtures account for most of the emission uptakes (~31 of 42 kg during EMeRGe-Europe and ~323 of ~454 kg during EMeRGe-Asia), and hence we can assume that they will dominate measured trace gas enhancements.

### 4.3.    Chemical fingerprints of source region emissions

By linking the modelled trajectory- and inventory-based partitioning of source regions and MPCs (Table 5, based on the approach explained in Fig. 3) and the observed VOC-based emission signatures (see section 4.1), we aim to characterise air-masses of sampled source region and MPC emissions. The approach is simple. We used the identified source region contributions (Table 5) as filter for our observed VOC-based signatures (Fig. 4b, c) to generate composites of the source region
emission signatures.

**EMeRGe-Europe**: Figure 10 shows the fractions of chemical fingerprints (emission signatures I and II, for a detailed explanation see Table 3) for the 27 source regions and MPCs listed in Table 5 (right). The main feature of the EMeRGe-Europe source regions are anthropogenic (AP, black) and background (BG, blue) signatures, whose fractions differ from source region to source region. The fraction of present background air reaches e.g. more than 50 % in LRT air-masses from USA and Canada
(CAN) as well as in the mixture of Iberian Peninsula (IBE) and Northern Africa (NAF). Here, VOCs are already decayed. Due to the CO uptake approach, measured CO might be decayed or diluted as well in these air-masses. When fractions of background air are small, anthropogenic signals dominate.

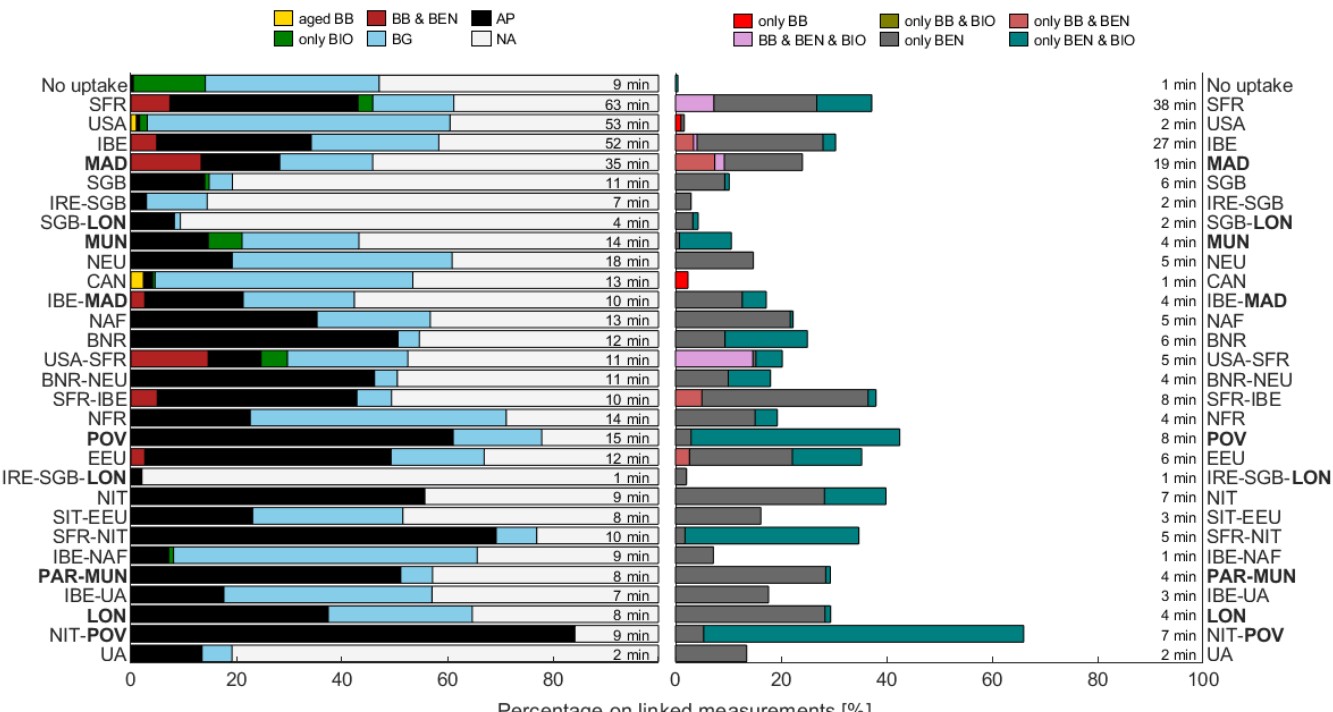

**Figure 10: Source region emission signatures (I - left, and II – right, using the tracer acetonitrile, benzene and isoprene) as chemical fingerprints during EMeRGe-Europe. AP – anthropogenic signatures, BB – biomass burning signatures, BEN – benzene enhancements, BIO – fresh biogenic signatures, BG – background (no tracer enhancement), NA – not assessable during instrumental background detection and during threshold transitions (due to PTR-MS measurement resolution of one minute per tracer). The temporal contribution of identified signatures to the measurements is shown on the right-hand side of the bars and differ from signatures I to II due to the usage of two and three tracers, respectively. For full names of source regions see Table 4.**

Largest anthropogenic fractions (50 – 80 %) show Po Valley (POV), Northern Italy (NIT), Eastern Europe (EEU) as well as BeNeRuhr (BNR) signatures and some mixtures which comprise emissions of these regions. Anthropogenic signals occur also together with biogenic signatures (only BEN & BIO, Fig. 10, right), especially in air-masses originating from Po Valley and Northern Italy, indicating the mixture of recent anthropogenic and biogenic emissions. Separate biogenic signals (only BIO) are sparse and only present in Munich (MUN), Southern France (SFR) as well as the mixture USA-SFR. Minor biogenic signals are surprisingly also present in USA and air-masses with no uptakes, probably due to mixed in local biogenic emissions not covered by the CO uptake scheme.

We could also identify BB signals in outflows of Southern France, the Iberian Peninsula, Madrid (MAD), and Eastern Europe (EEU), as well as in some mixtures of emission regions (e.g. USA-SFR and SFR-IBE). These BB events (identified by enhanced acetonitrile) were mostly rich in benzene too, indicating fresh burnings and/or mixtures with anthropogenic sources.

A minor signal of aged BB is present in air-masses influenced by Canadian emissions.



All three tracers show partly simultaneous enhancements in air-masses originating from Southern France and in the mix of Southern France/USA, which can be attributed to a local BB event near Marseille that was directly visible from the airplane during flight EU-07.

Due to i) instrumental background detection, ii) the PTR-MS duty cycle gaps (as described in section 3.1.2 and the supplement)
and iii) no available VOC data during flight EU-05 (targeting London), the possible assessment of emission signatures differs largely in the different source region composites, represented by the white bars in Fig. 10 (left).

**EMeRGe-Asia**: During this campaign part, we obtain chemical fingerprints of 25 source region composites (Fig. 11). In contrast to EMeRGe-Europe, the source region emissions show much larger fractions of anthropogenic signatures (black) and BB signatures (dark red and yellow). Largest anthropogenic fractions (60 – 90 %) are present in air-masses originating from
Northeastern China (NEC), Yangtze Delta (YAN), Eastern China (ECH), Taipei (TAI), the area Xian-Beijing-Shanghai (XBS) and mixtures of those regions. The pure anthropogenic signatures, occur mainly without other signatures. Only in some cases they are accompanied by biogenic signals, e.g. in Taiwan mixtures, most likely due to the sampling of local fresh emissions from the PBL.

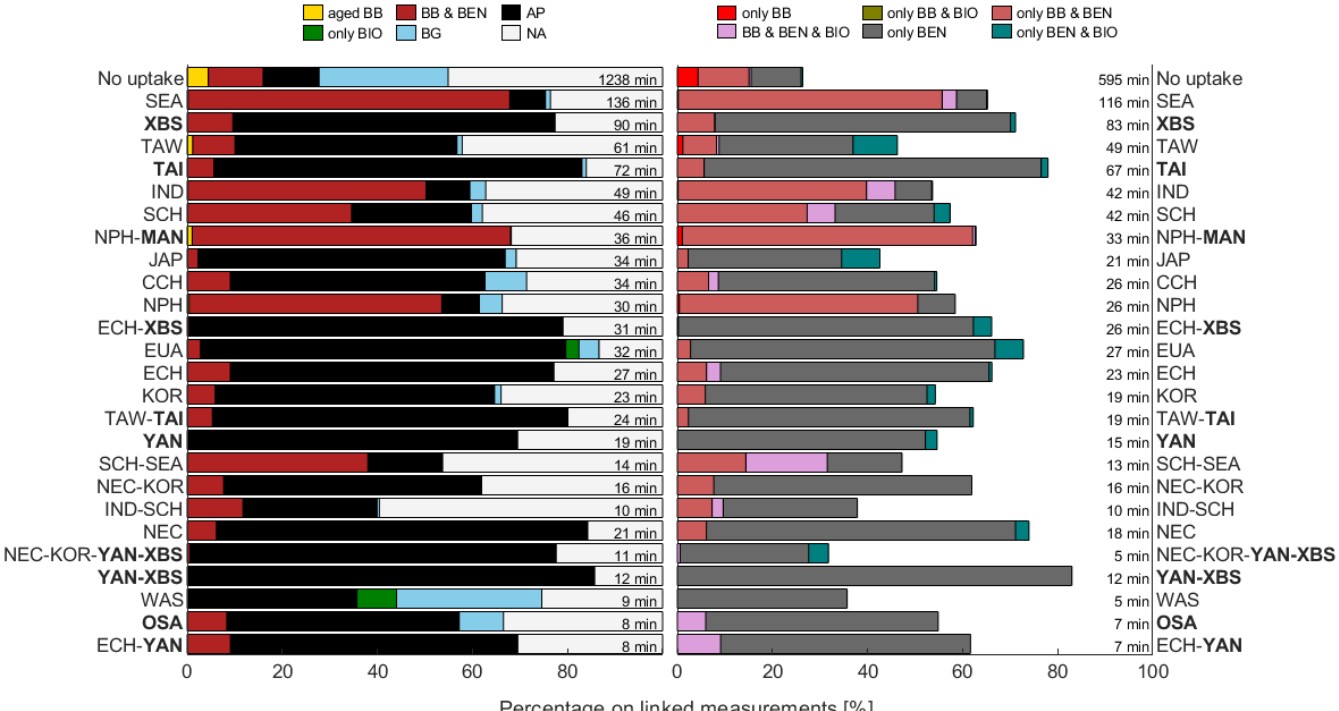

**Figure 11: Source region emission signatures (I - left, and II – right, using the tracer acetonitrile, benzene and isoprene) as chemical fingerprints during EMeRGe-Asia. AP – anthropogenic signatures, BB – biomass burning signatures, BEN – benzene enhancements, BIO – fresh biogenic signatures, BG – background (no tracer enhancement), NA – not assessable during instrumental background detection and during threshold transitions (due to PTR-MS measurement resolution of one minute per tracer). The temporal contribution of identified signatures is shown at the right-hand side of the bars and can differ from signatures I to II due to the usage**
**of two and three tracers, respectively. For full names of source regions see Table 4.**





We identified BB signals in nearly all outflows, besides Western Asia (WAS) and the Yangtze Delta (YAN) mixtures. However, Southeast Asia (SEA), the Northern Philippines (NPH) and the emission mixture with Manila (NPH-MAN), India (IND), and Southern China (SCH) show the largest BB influence (30 – 70 %). In air-masses without uptake, we identified also aged BB signals and hence processed air. Emission signatures II (Fig. 11, right) show that BB occurs mainly together with

benzene enhancements, indicating fresh BB events and/or mixtures with anthropogenic emissions.

The enhancement of all three tracers (pink bar, e.g. in Southeast Asian mixtures and India) may indicate fresh BB. However, why short-lived isoprene is still present in air-masses transported from India, requires further investigation. Same applies for air-masses from Western Asia (WAS) and Europe/Africa (EUA).

Unambiguously, the fraction of background air (BG) is quite small in the source region composites of EMeRGe-Asia (less

than 10 %); only air-masses with no uptake and from Western Asia show a larger fraction (~30 %) of no present VOC-based emission signatures.

## 5.     Summary and conclusion

In light of increasing urban agglomerations and global warming, the airborne campaign EMeRGe (2017/2018) investigated transport patterns and transformation processes of European and Asian megacity outflows. Using observed VOCs as chemical

tracers, back-trajectories and the EDGAR emission inventory, we characterise the air-masses probed during EMeRGe-Europe (July 2017) and EMeRGe-Asia (March/April 2018) and identify their origins, attribute emission properties and discuss the fate of the emissions while being transported towards our sampling region.

We measured up-to nine different VOCs simultaneously (see Table 1), which allowed the identification and characterization of polluted air-masses by the exceedance of certain VOC thresholds. During EMeRGe-Europe, air-masses with no

enhancements, that is, background air, dominated. However, anthropogenic signals (identified using benzene) were encountered frequently, but BB events (identified using acetonitrile) played a minor role. Furthermore, we identified minor fresh biogenic emissions during EMeRGe-Europe, partly coinciding with anthropogenic signals. During EMeRGe-Asia, fresh anthropogenic signals dominated, followed by frequent BB events, but rare encounters of fresh biogenic emissions.

To attribute sampled pollutants to their source regions, we seeded back-trajectories along the flightpath (using FLEXTRA,

driven by ERA5 winds) with CO emissions derived from the EDGAR emission inventory (see section 3.2). With this Lagrangian approach, we modelled the potential for CO emission uptake, when trajectories traverse the planetary boundary layer. For simplicity and due to the short back-trajectory duration of 10 days, we neglected loss of CO by oxidation and dilution processes during transport. The majority of all emissions (~40%) during EMeRGe-Europe originated from South Great Britain, the Belgium-Netherlands-Ruhr area and Southern France (see section 4.2.2. and Fig. 7). The seven target MPCs (listed in

Table 5) contributed to about 16.5 % of the modelled emissions. During EMeRGe-Asia, HALO probed to 66 % the outflow of Mainland China (see Fig. 8), with further contributions of ~25 % from Taiwan (12%), Korea (7 %) and Southeast Asia (6 %). Specifically, emissions of target MPCs (Table 5) contributed to ~37 %, mainly from the megacity agglomeration inside





the triangle Xian-Beijing-Shanghai and from some fresh emissions of Taipei and Manila. During both campaigns, both fresh and chemically processed air was sampled, in which ~50 % of the probed air-masses saw emissions in the last three to four
days prior to the measurement. Overall, our emission inventory-based approach indicated ~6 times higher amounts of pollutants at sampling altitudes (with comparable flight characteristics, see Fig. 1) during EMeRGe-Asia than during EMeRGe-Europe, with respect to the completed flight hours.

To analyse source region specific fingerprints, we linked the modelled anthropogenic CO uptakes with the EMeRGe observations. Since air parcels (represented by the trajectories) can traverse trough PBLs of multiple source regions, the
resulting air-masses sampled on board HALO may not only comprise traces of emissions from a single source region (referred to as non-mixed) but also of different regions (referred to as mixed). To consider only "significant emissions" for all uptake events, we omitted source regions with small uptake contributions (in sum 5 % of the total uptake sum of the respective campaign part, see section 4.2.3). In sum, we could link ~50 % (EMeRGe-Europe) and ~30 % (EMeRGe-Asia) of all sampled air-masses to certain emission regions and MPCs. The chemical fingerprints differ from region to region. During EMeRGe-
Europe, we found the largest fractions of anthropogenic signals in outflows of Po Valley, Northern Italy, London and the BeNeRuhr area, during EMeRGe-Asia of Northeast and East China as well as of Taipei. During EMeRGe-Europe, the anthropogenic signals coincide partly with biogenic signatures, indicating recent contact to the PBL. We identified BB signals in outflows of Southern France, the Iberian Peninsula, Madrid and Eastern Europe (EMeRGe-Europe) as well as of Southeast Asia, the Northern Philippines (including Manila), India, and Southern China (EMeRGe-Asia). The BB signals are almost
exclusively accompanied by benzene enhancements, indicating fresh burning events and/or mixtures with anthropogenic sources, since benzene can originate from both, BB and anthropogenic sources.

Overall, our "source attribution approach" showed i) that during most flights anthropogenic emissions of target regions have been successfully probed, and ii) that this approach in conjunction with the measured VOC-based (chemical) fingerprints provides reasonable findings. However, the majority of identified emissions could not be attributed to the target MPCs, because
of the large-scale measurement approach of EMeRGe and the numerous involved emission regions. Closer sampling in altitude and location to the MPCs was unfortunately not possible due to the strict flight restrictions near the target population hotspots. Nonetheless, the identified emission signatures and regions enable subsequent studies either to analyse other trace gas measurements in the different source regimes or to help refine the dynamical identification of individual MPC outflows. A more detailed characterisation of MPC outflows regarding chemical transformation is difficult because chemical and
microphysical transformations will lower concentrations such that relative enhancements become hard to detect. In addition, the dispersion and mixing of emission plumes at different processing stages make it very challenging to investigate the chemical transformations of individual MPCs in the measurements. Here, PFC tracer experiments conducted during EMeRGe can further support a comprehensive analysis.

In conclusion, EMeRGe provides unique near- and far-field trace gas data from emission hotspots to study the transport and
chemical processing of pollution outflows and to validate atmospheric models. To quantitatively capture a more robust picture of megacity emissions, a network of different measurement platforms with strong model support is required, which covers a





range of spatial and temporal scales: Local ground-based and local airborne measurements are necessary to understand the structure of emission outflows, large-scale airborne measurements provide information on the transport and mixture of multiple emission outflows and satellite measurements are essential to monitor global distributions, the evolution of outflows and their

processing into background air. Finally, model simulations enable the linkage of these versatile measurements.

With FLEXTRA back-trajectories based on ERA5 wind fields and the EDGAR emission inventory we utilised state-of-the-art model data together with an extensive set of VOC tracers enabling the study of elaborated chemical fingerprints. Hence, our work demonstrates the strength of combining the complementarity of a trajectory-based method to cross-reference regions of air-mass origin and direct measurements to highlight different emission sources/categories in describing MPC outflows

measured along flight routes, i.e. during EMeRGe. Finally, our approach provides an opportunity to test inventories and to improve underlying models.

**Appendix A**

**List of acronyms**

| | |
|---|---|
| AP | AnthroPogenic signatures |
| BB | Biomass Burning signatures |
| BIO | BIOgenic signatures |
| CityZen | megaCITY - Zoom for the ENvironment, https://cordis.europa.eu/project/id/212095 |
| E/N | Parameter used in ion mobility studies  E: electric field in the drift tube, N: number density of the gas in the drift tube, de Gouw et al., (2003) |
| EDGAR | Emission Database for Global Atmospheric Research |
| EMeRGe | Effect of MEgacities on the transport and transformation of pollutants on the Regional to Global scalEs |
| ERA5 | ECMWF (European Centre for Medium-Range Weather Forecasts) Reanalysis version 5 |
| FAAM | Facility for Airborne Atmospheric Measurements |
| FLEXTRA | FLEXible TRAjectory model |
| HALO | High Altitude and LOng range research aircraft |
| HKMS | HALO Karlsruhe Mass Spectrometer |
| LOD | Limit Of Detection |
| LRT | Long-Range Transport |
| MEGAPOLI | Megacities: Emissions, urban, regional and Global Atmospheric POLlution and climate effects, and Integrated tools for assessment and mitigation; 2009-2010, http://megapoli.dmi.dk |
| MILAGRO | Megacity Initiative: Local And Global Research Observations, Molina et al., (2010) |
| miniDOAS | Compact Differential Optical Absorption Spectrometer, Hüneke et al., (2017) |
| MPC | Major Population Centre |





| PBL | Planetary Boundary Layer |
| --- | --- |
| 725 PTR-MS | Proton-Transfer-Reaction Mass Spectrometry |
| Td | Townsend (1 Td = $10^{-17}$ V cm$^2$), Unit of E/N, de Gouw et al., (2003) |
| QMS | Quadrupole Mass Spectrometer |
| VMR | Volume Mixing Ratio |
| VOC | Volatile Organic Compound |


*Data availability.* VOC data of the HKMS are stored in the HALO database (https://doi.org/10.17616/R39Q0T, re3data.org, (2020)) and can be accessed upon signing a data protocol. EDGAR CO emission rates are available at https://edgar.jrc.ec.europa.eu/dataset_ap61. FLEXTRA back-trajectories merged with selected ERA5 parameters can be obtained upon request from APP (alepou@uni-bremen.de).


*Supplement.* The supplement of this article is available at:

*Author contributions.* EF operated the PTR-MS, processed the VOC raw data, analysed the data and prepared the manuscript with contributions and revisions of HB, MN, FO, AZ, APP, MV and PB. ML provided the CO measurements. AH, AKH, ND, 740 APP and MV calculated the FLEXTRA trajectories and merged them with ERA5 meteorological data. All authors have read and agreed to the published version of the manuscript.

*Competing interests.* The authors declare that they have no conflict of interest.

*Special issue statement.* This article is part of the special issue "Effect of Megacities on the Transport and Transformation of Pollutants on the Regional to Global Scales (EMeRGe) (ACP/AMT inter-journal SI)".

*Acknowledgements.* The authors thank the EMeRGe colleagues for their collaboration, helpful discussions and support during the HALO flight campaigns and numerous workshops. The FLEXTRA simulations were performed on the HPC cluster 750 "Aether" at the University of Bremen, financed by DFG within the scope of the Excellence Initiative.

Hersbach, H. et al., (2018) was downloaded from the Copernicus Climate Change Service (C3S) Climate Data Store. The results contain modified Copernicus Climate Change Service information 2020. Neither the European Commission nor ECMWF is responsible for any use that may be made of the Copernicus information or data it contains.


*Financial support.* The authors acknowledge the Deutsche Forschungsgemeinschaft (DFG) for funding this work within the project "Chemical composition and transformation of VOCs in the outflow of large population centers in Europe and Asia



during EMeRGe" (ChoColate, NE 2150/1-1) and the further support of the Karlsruhe Institute of Technology. ND, APP and MV acknowledge financial support by the Deutsche Forschungsgemeinschaft (DFG, German Research Foundation) under

Germany´s Excellence Strategy (University Allowance, EXC 2077, University of Bremen) and the University of Bremen.

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
