# Peer review of "Chemical and dynamical identification of emission outflows during the HALO campaign EMeRGe in Europe and Asia"

_Atmospheric Chemistry and Physics, 2022_

## Author Response (AR1)

**Author response on anonymous referee #1 (R1), acp-2022-455**

*The manuscript by Förster et al. provides an insight into measurements conducted as part of the EMeRGe campaign. The analysis is thorough and convincing. I recommend publication after addressing comments below.*

**Answer:** First of all, we would like to thank R1 for carefully reading the manuscript and for the generally positive, helpful comments. We have followed all suggestions for change and have modified the manuscript and Supplementary Information accordingly. Below we provide point by point responses to the *comments* with given lines of changes (if applicable) and highlighted the corresponding changes made in the manuscript (additional document) in yellow.

**Author response**

*Define MPC – what size, what criterium?*

**Answer:** We specified the definition in lines 63-65: "…, we will use the term MPC to describe megacities with more than 10 million inhabitants as well as metropolitan areas or converging urban conurbations with more than four million inhabitants (e.g. metropolitan area of Rome or Madrid)."
However, like the definition of megacity (the only criterion is more than 10 million inhabitants) the term MPC can just be loosely defined.

*Line 110- 120: The authors mention emissions extensively throughout the manuscript, however they do not quantify nor measure emissions directly. Typically emissions as opposed to concentrations are defined as g/s or g/m2/s or similar. Whenever appropriate I suggest to rephrase these statements. The overarching measurement approach was not designed to specifically quantify emissions (a typical aircraft setup for monitoring emissions for example: Yuan et al., 2015 (doi: 10.1002/2015JD023242 and Baray et al., 2018: doi: 10.5194/acp-18-7361-2018). Rather I would call the measurement approach to characterize and interpret atmospheric concentrations (e.g. aged composition rather than aged emissions). What is actually promoted here is the use of the chemical composition of certain tracer VOC as a relative fingerprint of pollution sources.*

**Answer:** We agree that the term "emission" was used in two ways throughout the manuscript: First as term for a general description of "release of trace gases" from various sources, e.g. biogenic or anthropogenic, without a quantitative statement. And second, as modeled CO emission uptakes, with a quantitative indication.

For the first case (related to measured trace gas enhancements), we mostly changed the term "emission" to "pollution" or "composition" throughout the manuscript as suggested by R1. We added also a short paragraph defining pollution as emission impact (lines 101-105):

"In general, localised (intensive) trace gas emissions of urban areas rapidly increase corresponding volume mixing ratios in affected air masses. Chemical conversion and dilution can attenuate such emission driven enhancements of tracer volume mixing ratios in a given air mass. However, in most cases emission impacts can be clearly detected by measuring relative enhancements of volume mixing ratios and here we will to refer to such enhancements as pollution (events)."

Since the modelled emission uptake is given as a mass, we kept the term for the modelling of the anthropogenic CO uptake. However, we tried to be more precisely and using now mostly the terms "emission uptake" or "modelled emission" throughout the modelling discussion.

*Line 151: this is true on a global scale, however in the northern hemisphere the anthropogenic contribution is much larger (up to 50%). This has also been shown recently by a number of VOC measurements in urban areas (e.g. Karl et al., 2018, doi: 10.1073/pnas.1714715115; Gkatzelis et al., 2021: doi : 10.1021/acs.est.0c05471). The regional distribution of VOC should be discussed in this manuscript.*

**Answer:** We thank R1 for the remark. The text has been modified accordingly (lines 158-163):
"Globally, the vast majority of VOCs in the atmosphere are emitted by biogenic sources (Guenther et al., 1995; Sindelarova et al., 2014), thus from vegetation or biomass fires (Ciccioli et al., 2014). However, the regions with the largest biogenic emissions are the rain forests in South America and Africa. In the northern hemisphere, biogenic VOCs are only emitted during the vegetation phase with overall smaller emissions rates (Guenther et al., 1995). Accordingly, recent urban measurements indicate that anthropogenic emissions account for about half of the VOC flux into the atmosphere of the northern hemisphere (Karl et al., 2018). Inside urban agglomerations, anthropogenic VOC emissions will even dominate (Amodio et al., 2013)."

*Line 166: for E/N of 142 I would expect that a large fraction on the 'benzene' mass can potentially originates from fragmentation of higher aromatics (e.g. ethylbenzenes) – the authors should give constraints on this interference*

**Answer:** This remark is now included in the text (lines 187-191):
"Furthermore, the mass of benzene (m/z=79) can potentially experience signals from the fragmentation of higher aromatics, e.g. from C8 aromatics like ethylbenzenes (m/z=107), due to the operation of the HKMS with a relatively high E/N value (de

Gouw and Warneke, 2007). Nevertheless, higher aromatics originate mostly from anthropogenic sources as well, why we consider the signal of m/z = 79 still as suitable tracer for anthropogenic activities."

*Line 194: Acetonitrile has been used as a biomass burning marker, however the magnitude of acetonitrile and HCN emissions depends on the fuel N content – urban burning events for example are often not captured adequately*

**Answer:** This remark is now included in the text (lines 185-187):

"However, it should be noted, that the magnitude of acetonitrile emissions depends on the nitrogen content of the burned fuel, showing lower concentrations in residential wood burning and hence, acetonitrile may not be a suitable tracer for domestic burning in urban areas (Coggon et al., 2016)"

*Line 201: It should be noted that there are also isoprene (and furan) emissions from biomass burning, which could dilute the specificity of this tracer.*

**Answer:** Therefore, we consider for the source signature "only biogenic" just isoprene enhancements without the simultaneous enhancement of acetonitrile and benzene, to filter out possible fresh BB sources of isoprene. We try to make it clearer in the
manuscript (lines 220-221):

"This source signature denotes only a biogenic signal without enhancements of benzene and acetonitrile (filtering out BB induced isoprene emissions, e.g. Müller et al., 2016)."

*Line 240: emissions from the RCP (residential, commercial and public) as well as industrial sector do not only happen during daytime. The analysis of FLEXPART could therefore suffer from that assumption.*

**Answer:** The text has been accordingly modified to consider the nighttime emissions and the possible influence on the analysis due to higher PBL uncertainties (lines 267-268):
"However, emissions from the RCP (residential, commercial and public) as well as industrial sector can also arise during the night, where larger PBL height uncertainties might have an influence on our analysis."

*Line 289: I understand the influence of chemistry here for the degradation of CO, but not transport- Shouldn't Flexpart handle dilution as well*

**Answer:** For our analysis, we did not use the dispersion model FLEXPART itself but "only" the trajectory model FLEXTRA (that does not handle dilution). The main reason why we used FLEXTRA (+ ERA5 PBL + EDGAR) is that most of our measurements (80%) are outside the PBL and nearly half of the probed air masses (45%) haven't been in the PBL (where the emissions are) during the last 10 days. In the free troposphere above the PBL "simple trajectories" are sufficient to identify the mean transport pathways, mainly due to less turbulence/shear compared to the PBL.

Source-receptor-relations between emission sources at the ground and airborne observations can only be calculated by dispersion models like FLEXPART. This would be needed to quantify source contributions of pollution plumes, i.e. calculating the enhancement of CO mixing ratios by the different sources (at the ground) at the location of the aircraft. However, this quantification is not the intention and beyond the scope of this paper. Our intention is to identify and characterize in a first step all air masses probed during EMeRGe in an efficient way. The results allow then further analysis, e.g. dispersion modelling for specific cases.

We added a short paragraph in the manuscript for clarification (lines 256-262).

*Fig4. I would rather call this a source signature rather than an emission signature. Background air is NOT an emission signature – this refers again to my comment at the beginning -> I would be more concise about the actual meaning and definition of emissions throughout the manuscript.*

**Answer:** The term "emission signatures" was changed to "source signatures" throughout the manuscript. We agree that this is a more suitable term.

*Section 3: The authors classify aged and non aged emissions. To that end the ratio of Toluene/benzene could help to verify their assumptions for airmasses without a large influence of biomass burning. Do the aged ratios agree with the estimated calculations from back trajectories assuming typical OH densities (e.g. from CMIP climatology models)?*

**Answer:** The ratio of TOL/BEN was investigated in earlier data analyses to estimate air mass ages over a chemical clock. Unfortunately, due to the lack of ratios near the emission sources (climatological emissions does not help here, because we are looking mostly at specific events) and the strong mixture of different anthropogenic sources (with different air mass ages), the clocks lead to no clear age estimation. Therefore, the ratio of TOL/BEN is for EMeRGe in general not suitable to verify the transport time given by the trajectories. However, with the TOL/BEN ratios itself, fresh and aged air masses have been estimated for plumes with toluene enhancements in an earlier analysis (Förster et al., 2018).

**References**

Amodio, M., de Gennaro, G., Marzocca, A., Trizio, L. and Tutino, M.: Assessment of Impacts Produced by Anthropogenic Sources in a Little City near an Important Industrial Area (Modugno, Southern Italy), Sci. World J., 2013, 1–10, doi:10.1155/2013/150397, 2013.

Ciccioli, P., Centritto, M. and Loreto, F.: Biogenic volatile organic compound emissions from vegetation fires, Plant. Cell Environ., 37(8), 1810–1825, doi:10.1111/pce.12336, 2014.

Coggon, M. M., Veres, P. R., Yuan, B., Koss, A., Warneke, C., Gilman, J. B., Lerner, B. M., Peischl, J., Aikin, K. C., Stockwell, C. E., Hatch, L. E., Ryerson, T. B., Roberts, J. M., Yokelson, R. J. and de Gouw, J. A.: Emissions of nitrogen-containing
organic compounds from the burning of herbaceous and arboraceous biomass: Fuel composition dependence and the variability of commonly used nitrile tracers, Geophys. Res. Lett., 43(18), 9903–9912, doi:10.1002/2016GL070562, 2016.

Förster, E., Neumaier, M., Obersteiner, F., Bönisch, H. and Zahn, A.: First results of VOC and ozone measurements in European and Asian Major Population Centers (MPC) during the research aircraft campaign EMeRGe (2017/2018), in
IGAC/iCACGP conference, Takamatsu, Japan. [online] Available from: https://www.researchgate.net/publication/327980909_First_results_of_VOC_and_ozone_measurements_in_European_and_Asian_Major_Population_Centers_MPC_during_the_research_aircraft_campaign_EMeRGe_20172018, 2018.

de Gouw, J. and Warneke, C.: Measurements of volatile organic compounds in the earth's atmosphere using proton-transfer-
reaction mass spectrometry, Mass Spectrom. Rev., 26(2), 223–257, doi:10.1002/mas.20119, 2007.

Guenther, A., Hewitt, C. N., Erickson, D., Fall, R., Geron, C., Graedel, T., Harley, P., Klinger, L., Lerdau, M., Mckay, W. A., Pierce, T., Scholes, B., Steinbrecher, R., Tallamraju, R., Taylor, J. and Zimmerman, P.: A global model of natural volatile organic compound emissions, J. Geophys. Res., 100(D5), 8873, doi:10.1029/94JD02950, 1995.

Karl, T., Striednig, M., Graus, M., Hammerle, A. and Wohlfahrt, G.: Urban flux measurements reveal a large pool of oxygenated volatile organic compound emissions, Proc. Natl. Acad. Sci. U. S. A., 115(6), 1186–1191, doi:10.1073/pnas.1714715115, 2018.

Müller, M., Anderson, B. E., Beyersdorf, A. J., Crawford, J. H., Diskin, G. S., Eichler, P., Fried, A., Keutsch, F. N., Mikoviny, T., Thornhill, K. L., Walega, J. G., Weinheimer, A. J., Yang, M., Yokelson, R. J. and Wisthaler, A.: In situ measurements and modeling of reactive trace gases in a small biomass burning plume, Atmos. Chem. Phys., 16(6), 3813–3824, doi:10.5194/acp-16-3813-2016, 2016.

Sindelarova, K., Granier, C., Bouarar, I., Guenther, A., Tilmes, S., Stavrakou, T., Müller, J.-F., Kuhn, U., Stefani, P. and Knorr, W.: Global data set of biogenic VOC emissions calculated by the MEGAN model over the last 30 years, Atmos. Chem. Phys., 14(17), 9317–9341, doi:10.5194/acp-14-9317-2014, 2014.

**Author response on anonymous referee #2 (R2), acp-2022-455**

*Review of Chemical and dynamical identification of emission outflows during the HALO campaign EMeRGe in Europe and Asia, by Förster et al.*

*This manuscript details the use of observations of a few specific VOCs made during two HALO flight deployments to identify the influence of major source types by using threshold values of compounds to identify influence of biomass burning emissions, anthropogenic emissions, biogenic emissions, or combinations of the above. The authors then use back trajectory modeling to attribute and quantify these various contributions back to specific regions.*

**Answer:** First of all, we would like to thank R2 for carefully reading the manuscript and for the generally positive, helpful comments as well as the detailed technical corrections. We have followed all suggestions for change and have modified the manuscript and Supplementary Information accordingly. Below we provide point by point responses to the *comments* with given lines of changes (if applicable) and highlighted the corresponding changes made in the manuscript (additional document)

in blue.

**Author response**

*The presentation of the research is a little tedious to get through, but in general, it seems like a reasonable methodology for broadly assessing air masses sampled at varying distances from their likely sources. I have a few issues with the exact approach, however. One major issue I have is that the authors use in their back trajectory modeling to the PBL is that it discounts biomass burning injection height, and lofting due to convection, which can be significant over the Tibetan Plateau*

*due to the Asian Monsoon. The authors should address this possible contribution and implications in their in their work.*

**Answer:**

We agree that lofting by convection is an export mechanism of emissions, especially in the tropics. Since we i) only wanted to set up a simple model and ii) most of the probed emissions emanate from mid-latitude sources (here low-level advection dominates according to Folberth et al., 2015), we have decided to neglect lofting by convection. We added a paragraph (lines 292-296) describing this point.

We also agree that over the Tibetan Plateau convection is a significant transport mechanism. However, in the case of EMeRGe-Asia, the effect of the Tibetan Plateau can be neglected because i) we measured in an inter-monsoon period (March/April) and ii) there are nearly no emissions of BB and anthropogenic sources over this region that could be transported. So the impact would be negligible.

A biomass burning injection height is not applicable here because we only consider anthropogenic CO in the trajectory model. We added a clarification in the introduction of Sect. 3.2.2. (lines 290-291).

*Secondly, the authors appear to put far more emphasis on the value of the back trajectory modeling than in the VOC tracers that they are studying.*

**Answer:** Due to the multiple targets of EMeRGe and the large-scale probing, the dynamical identification of pollution outflows (back trajectory modeling) is an important part of the campaign analysis to determine/estimate the overall source contributions during EMeRGe. The VOC tracers alone are a helpful tool to characterise air masses, but in the case of EMeRGe the larger emphasis on the modelling was necessary to identify the origin of the probed air masses, and hence to characterise specific pollution outflows of target areas.

*Lastly, due to the nature of aircraft campaigns, there is an inherent sampling bias to air masses from specific source regions, and that should also be addressed in the paper.*

**Answer:** The issue of a sampling bias is now addressed in the manuscript (lines 137-138). We want also to emphasise that we did not make any conclusions, which can be invalidated due to this sampling bias.

***Some more specific comments follow, with a large number of technical corrections below.***

**Answer:** We thank R2 for the detailed specific comments, which have all been implemented.

*Line 37 – consider using "rural areas" instead of "countryside", and perhaps consider that in many countries there is a suburban interface, and identify whether that is "urban" or "countryside/rural" in the UN statistics.*

**Answer:** Changed to rural areas (line 37). The UN (United Nations, Department of Economic and Social Affairs, Population Division, 2018) uses the three concepts "city proper", "urban agglomeration" and "metropolitan area":

"… "city proper", describes a city according to an administrative boundary. A second approach, termed the "urban agglomeration", considers the extent of the contiguous urban area, or built-up area, to delineate the city's boundaries. A third concept of the city, the "metropolitan area", defines its boundaries according to the degree of economic and social interconnectedness of nearby areas, identified by interlinked commerce or commuting patterns, for example."

We would see the suburban interface as additional area between urban agglomeration and metropolitan area, which is included by the UN for 10 % of the evaluated cities:

"The 2018 revision of World Urbanization Prospects (WUP) endeavoured wherever possible, given available data, to adhere to the "urban agglomeration" concept of cities. Often, however, in order to compile a series of population estimates that was consistent for a city over time, the "city proper" or "metropolitan area" concepts were used instead. Of the 1,860 cities with at least 300,000 inhabitants in 2018 included in WUP, 55 per cent follow the "urban agglomeration" statistical concept, 35 per cent follow the "city proper" concept and the remaining 10 per cent refer to "metropolitan". (United Nations, Department of Economic and Social Affairs, Population Division, 2018)

*Line 75 – "(and that provides only one time information based on its lifetime…" is not communicating what the authors are trying to say.*

**Answer:** The text was adapted to "Such an analysis is advantageous compared to the conventional approach of using a single tracer like carbon monoxide (CO) with its rather long lifetime (~2 months) and multiple sources (anthropogenic and biomass burning)." (Lines 76-77)

*Lines 115-116 – "some 100 km" – from what?,*
**Answer:** Changed to "100 km downstream of targeted source regions". (line 120)

*and on the next line, perhaps "source" instead of "target"? since the regions are where the emissions are coming from, not heading to.*
**Answer:** Changed to "targeted source region". (line 120)

*Also, "concept is mirrored by"… mirrored doesn't seem like the right word here.*
**Answer:** Changed to "reflected". (line 121)

*Also, one might argue that at 100 km, emissions are not exactly fresh.*

**Answer:** Changed to "recently emitted". (line 120)

*The other reviewer makes very good points about emissions vs. characterizing source concentrations, so I won't belabor that*
*point, but I will echo that it needs to be addressed throughout.*

**Answer:** We agree that the term "emission" was used in two ways throughout the manuscript: First as term for a general description of "release of trace gases" from various sources, e.g. biogenic or anthropogenic, without a quantitative statement. And second, as modeled CO emission uptakes, with a quantitative indication.

For the first case (the measurements), we mostly changed the term "emission" to "pollution" or "composition" throughout the
manuscript as suggested by R1. We added also a short paragraph defining pollution as emission impact (lines).

Since the modelled emission uptake is given as a mass, we kept the term for the modelling of the anthropogenic CO uptake. However, we tried to be more precisely and using now mostly the terms "emission uptake" or "modelled emission" throughout the modelling discussion.

*Line 162 – "to the scientific needs" requires more explanation.*

**Answer:** We added a more detailed explanation (lines 173-174): "… scientific needs, e.g. customisable duty cycle of measurements/background detection due to meteorological conditions/ conditions of aircraft campaign and full access to all
operating parameters."

*Table 3 – is there a reason why some columns are italicized and some aren't? This isn't clear.*

**Answer:** We thank R2 for the note. Originally, the italicized columns should designate the source signatures I with the corresponding VOC enhancements (or no enhancement) of acetonitrile and benzene used for the definition. We have removed italics from all columns to avoid confusion.

*Lines 223-225 – it is also not clear to me how nine consecutive 6-s measurements can be reasonably interpolated to 1-s data.*

**Answer:** We thank R2 for the remark. Actually, we did not interpolate the VOC measurements itself to 1s, but extended the assignment of source signatures (based on the measurements of three VOCs) to the general measurement frequency of 1s. We rephrased the text in the manuscript (lines 240-242) and in the supplement:

"Due to the PTR-MS duty cycle (consecutive integration of nine VOCs with ~6 s each), we assign the identified source signatures to the general measurement frequency of one second to enlarge the data coverage (see supplementary Sect. S4 for detailed description)."

*Line 251 – replace reddish colours with "red", as the high emissions are just one colour of red.*

**Answer:** Text was adapted (line 278).

*Figure 2 – these plots are very difficult to read, and are far too busy. Perhaps reduce the intensity of the background colours so that the text, flight paths, etc. can contrast against the background.*

**Answer:** The colour scheme of Figure 2 was adapted.

*Lines 562-563 – In 19% and 34% of what are negligible contributions inferred? Be specific.*

**Answer:** Text was adapted (lines 596-597): Moreover, in 19% (EMeRGe-Europe) and 34% (EMeRGe-Asia) of the total flight time small/negligible contributions are inferred and not considered.

*Technical corrections:*

**Answer:** We thank R2 for the detailed technical corrections which have all been implemented.

*Throughout A: "air-mass" or "air-masses" should be changed to "air mass" or "air masses".*

*Throughout B: remove spaces before % signs (e.g., line 28: "20 %" should be "20%".)*

*Throughout C: refer to style guide on referencing sections, specifically "The abbreviation "Sect." should be used when it appears in running text and should be followed by a number unless it comes at the beginning of a sentence."*

*Throughout D: "back trajectories" doesn't have a hyphen.*

*Throughout E. Be consistent with "up-take" vs. "uptakes" – the latter of which is correct (e.g., Table S4)*

*Line 38 – "the majority of megacities are still…"*

*Line 39 – what is meant by "extension"? Maybe not the right word.*

**Answer:** The spatial extent of a city. Changed to "dimension". (line 39)

*Line 48 – "… focused, e.g., on emissions…" (add commas)*

*Line 50 – there appears to be an extra space after "2008-2011"*

*Line 54 – "(Andrés Hernández et al., 2000 and references therein)" (remove inner parentheses)*

*Line 62 – "refer to Andrés Hernández et al. (2000)." (no comma)*

*Line 83 – "joined" should be "joint"*

*Line 119 – define ERA5*

*Line 123 – instead of "local burning", use "local fire".*

*Line 125 – for simplicity, use "chemical age, i.e., chemically old…" rather than "that is"*

*Line 129 – "starting from" (not form)*

*Line 135 – "10-day FLEXTRA", "back trajectories" (no hyphen).*

*Line 146 – remove the space before H3O+.*

    *Line 151 – "vast majority of VOCs in the atmosphere are" (not is)*

    *Line 152 – delete "further"*

    *Line 156 – "nighttime"*

    *Line 157 – "lifetimes"*

*Line 162 – "allows adaptations"*

    *Line 164 – limit of detection = LOD, lower detection limit = LDL? Maybe use "lower limit of detection (LOD)"? Also, "in the pptV range". (no hyphen)*

**Answer:** Changed to "limit of detection (LOD) in the lower pptV range". (line 175-176)

    *Lines 168-170 – "tropospheric lifetime t", and again, "lower limit of detection (LOD…" change "and the up-to four atmospheric main sources" to "… and the up to top four main atmospheric sources."*

*Line 178 – maybe "as explained below."*

    *Line 181 – "lifetime"*

    *Lines 189-190 – "please see details in the supplement" – perhaps instead refer to a specific supplement*

    *Line 205 – "… VOC tracers are…"*

    *Line 218 – "lifetimes"*

*Line 225 – again, perhaps point to a specific section in the Supplement.*

    *Line 235 – "10-day back trajectories…"*

*Line 236 – "the time step…" (no hyphen)*

*Line 251 – replace reddish colours with "red", as the high emissions are just one colour of red.*

*Line 265 – delete "Earth's" (it is redundant.) and perhaps use "trajectory air parcel" instead of air mass.*

*Line 266 – recommend changing to "… emissions rates at that location."*

*Line 272 – replace "exemplary" with "example"*

*Line 276 – immerses is not the right word here. Even "dips" would be better, or "drops", "descends", likely best.*

*Line 291 – define AMTEX*

*Line 304 – "… values listed in Table S2)." (S indicates supplement, so "in the supplement" is unnecessary).*

*Line 357 – "Figures 5 and 6…"*

*Line 365 – "the blue dot" is very difficult to see. Again – I recommend if possible having a less intense colour scheme for the background colour bar information, so that the details on top can be seen. Same comment for Figure 6.*

**Answer:** The colour scheme of Figure 5 and 6 has been adapted accordingly to Figure 2.

*Line 381 – "some hot spots such as the…", also include a comma after Madrid.*

*Line 385 – "More regions have likely…" – begs the question "more than what"? Be specific.*

**Answer:** Adapted the text: "Compared to EMeRGe-Europa, more regions have likely contributed … " (line 419)

*Line 397 – "Coordinates are shown in Table S3." Is sufficient.*

*Line 402 – "Figures 7 and 8…", also, Line 405 – "Figs. 7 and 8…"*

*Line 411 – BeNeRuhr needs to be defined. And consider whether two short forms (BNR as well) might add confusion.*

**Answer:** Now only BNR is used as abbreviation throughout the manuscript.

*The text in Figures 7 and 8 in the blue boxes is really difficult to read. Also, for both captions, "(plain) is not a great description. Perhaps use "the last row in each section", or "the uncoloured row".*

**Answer:** The colour scheme of Figure 7 and 8 has been adapted accordingly to Figure 2, 5 and 6. "Plain" was changed to
"uncoloured rows".

*Line 530 – "20 July 2017".*

*Line 604 – "fresh biomass burning" or "fresh fires"*

***Supplement***

*Line 3 and Table S1 – "Takeoff" and "Departure" are the same thing. I think you mean Takeoff and Landing, or Departure and Arrival. Just a note, as well – this may be the convention for HALO flights, but using ICARTT nomenclature, the flight*
*date is always the UTC departure/take-off date, regardless of how close it is to midnight, UTC.*

**Answer:** We changed the dates to the departure/take-off date. This is also the convention for HALO flights; however, we originally used the two "shifted" dates internally.

*Line 10 – "measurement noise by"*

*Line 17 – "4 April 2018"*

*Line 19 – Instead of writing "To Section 3.1.2, give the various sections in the supplement Section numbers (e.g., Sect. S1,*
*Sect. S2.4, etc.), which will make reader navitation between the main manuscript and the supplement much more straightforward.*

*Line 42 – "Same applies for isoprene." Is not a complete sentence.*

**Answer:** Added isoprene to the first sentence ("…, where the tracers acetonitrile, isoprene and benzene are measured …"). (line 33)

*Line 43 – "This assumption…" or "These assumptions…"*

*Line 45 – "as an additional tracer…"*

*Line 48 – this is far too much prose for a Table title. Move most of this to a text paragraph or to table footnotes. Also, the last line "This figure is a supplement…." This isn't a figure - it's a table. Also, I recommend only putting "Table X" or "Figure X" in bold, and not the entire title or caption.*

**Answer:** The table caption has been shortened and/or moved to a footnote (we shortened the caption of Figure 4, 10 and 11 in the main text accordingly). The entire caption is bold by default in the ACP MS Word template. In a published version, it will be only the label and not the whole caption, which, we agree, is indeed better.

*Line 58 – "MPCs (in italics) with coordinates…" or "MPCs (italicized) with coordinates…"*

*Lines 64 and 66 – delete "… in the main document." (it is unneeded, as sections and tables that don't begin with S are by definition in the main documents.*

**References**

Folberth, G. A., Butler, T. M., Collins, W. J. and Rumbold, S. T.: Megacities and climate change - A brief overview, Environ. Pollut., 203, 235–242, doi:10.1016/j.envpol.2014.09.004, 2015.

United Nations, Department of Economic and Social Affairs, Population Division: The World's Cities in 2018, World's Cities 2018—Data Bookl. (ST/ESA/ SER.A/417), 34, 2018.